**Effectiveness of emission control to reduce PM$_{2.5}$ pollution of Central China**
**during winter haze episodes under various potential synoptic controls**
Yingying Yan [1#], Yue Zhou [2#], Shaofei Kong [1, 4*], Jintai Lin [3], Jian Wu[1, 4], Huang Zheng
[1, 4], Zexuan Zhang [1, 4], Aili Song [1], Yongqing Bai [2], Zhang Ling [2], Dantong Liu [5],
Tianliang Zhao [6]
[1] Department of Atmospheric Sciences, School of Environmental Studies, China
University of Geosciences, Wuhan, 430074, China
[2] Hubei Key Laboratory for Heavy Rain Monitoring and Warning Research, Institute
of Heavy Rain, China Meteorological Administration, Wuhan 430205, China
[3] Laboratory for Climate and Ocean-Atmosphere Studies, Department of Atmospheric
and Oceanic Sciences, School of Physics, Peking University, Beijing 100871, China
[4] Department of Environmental Science and Engineering, School of Environmental
Studies, China University of Geosciences, Wuhan, 430074, China
[5] Department of Atmospheric Sciences, School of Earth Sciences, Zhejiang University,
Hangzhou, Zhejiang, China
[6] School of Atmospheric Physics, Nanjing University of Information Science and
Technology, Nanjing, 210044, China
*Correspondence to: Shaofei Kong ([kongshaofei@cug.edu.cn](mailto:kongshaofei@cug.edu.cn))*
[#] Contributed equally to this work
**Abstract**
Currently mitigating the severe particle pollution in autumn and winter is the key
to further improve the air quality of China. The source contributions and transboundary
transport of fine particles (PM$_{2.5}$) in pollution episodes are closely related to large-scale
or synoptic-scale atmospheric circulation. Under different synoptic conditions, how to
effectively reduce emissions to control haze pollution is rarely reported. In this study,
we classify the synoptic conditions over Central China from 2013 to 2018 by using
Lamb-Jenkension method and the NCEP/NCAR FNL operational global analysis data.
The effectiveness of emission control to reduce $PM_{2.5}$ pollution during winter haze
episodes under potential synoptic controls is simulated by GEOS-Chem model. Among
the ten identified synoptic patterns, four types account for 87% of the total pollution
days. Two typical synoptic modes of them are characterized by small surface wind
speed and stable weather conditions/high relative humidity (A/C-type) over Central
China due to a high-pressure system/a southwest trough low-pressure system, blocking
pollutants dispersion. Sensitivity simulations show that these two heavy pollution
processes are mainly contributed by local emission sources with ~82% for A-type and
~85% for C-type, respectively. The other two patterns lead to pollution of transport
characteristics affected by northerly/southerly winds (NW/SW-type), carrying air
pollution from northern/southern China to Central China. The contribution of pollution
transmission from North/South China is 36.9%/7.6% of $PM_{2.5}$ and local emission
sources contribute 41%/69%. We also estimate the effectiveness of emission reduction
in these four typical severe pollution synoptic processes. By only reducing $SO_2$ and
$NO_x$ emission and not controlling $NH_3$, the enhanced nitrate counteracts the effect of
sulfate reduction on $PM_{2.5}$ mitigations, with less than 4% decrease in $PM_{2.5}$. In addition,
to effectively mitigate haze pollution of NW/SW-type synoptic controlled episodes,
local emission control actions should be in coordination with regional collaborative
actions.

**1 Introduction**
The regional pollution of fine particles ($PM_{2.5}$) has attracted worldwide attention
in the public and in the scientific community (Cheng et al., 2016; Li et al., 2017c; Lin
et al., 2018; Bi et al., 2019) due to its detrimental effect on visibility (Wang et al., 2020)
and public health (Agarwal et al., 2017; Zhang et al., 2017). The $PM_{2.5}$ pollution in
China has been continuously alleviating since 2013 as the implication of the Air
Pollution Prevention and Control Action Plan (Zheng et al., 2018; Zhang et al., 2019),
especially in the Beijing-Tianjin-Hebei region (BTH) (Li et al., 2017b; Cheng et al.,
2019), the Yangtze River Delta (YRD) region and the Pearl River Delta (PRD) region.
However, severe particle pollution still occurs frequently in autumn and winter, which
is the major reason restricting the $PM_{2.5}$ to come up to national standard. For example,
12 extremely severe and persistent $PM_{2.5}$ pollution episodes occurred in Beijing in
January 2013, February 2014, December 2015, December 2016 and January 2017
(Zhong et al., 2018; Sun et al., 2016; Wang et al., 2018). Currently, how to effectively
reduce emissions in autumn and winter is the key to mitigate haze pollution in China.

The contribution of emission sources has been widely recognized as the decisive

factor of $PM_{2.5}$ pollution over urban agglomerations, including industrial exhaust, urban
transportation, residential emission, power plants, agricultural activities, and bio-
combustion (Huang et al., 2014; Tian et al., 2016; Wu et al., 2018; An et al., 2019).
While the outbreak, persistence and dissipation of particle pollution generally depends
on the meteorological conditions and regional synoptic patterns, controlled by the large-
scale or synoptic-scale atmospheric circulation (Chuang et al., 2008; Zhang et al., 2012;
Russo et al., 2014; Zheng et al., 2015; Shu et al., 2017; Li et al., 2019).

Many studies have tried to reveal the relationship between synoptic patterns and

severe particle pollution, and estimate the meteorological contributions to these
pollution episodes. The YRD is mainly affected by pollutants transmitted from the
northern and the southern China when the East Asian major trough is located at its front
(Liao et al., 2017; Shu et al., 2017; Li et al., 2019). Liao et al. (2020) has confirmed that
the relative position of the PRD to high-pressure systems imposes significant impacts
on the diffusion conditions and the $PM_{2.5}$ distributions in the PRD region. For North
China Plain (NCP), high frequency of stagnant weather accompanied by small pressure
gradient and near-surface wind speed, and shallow mixing layer are major reasons of
aerosol pollution over this region in winter (He et al., 2018). The aerosol pollution
formation process in Sichuan Basin is often controlled by the large scale high-pressure
circulation at sea level (Sun et al., 2020). In the Guanzhong basin, pollution event is
generally governed by both the large-scale synoptic situation and the small-scale local
circulation. The downhill wind not only forms a convergence zone in the basin, but also
makes pollutants flow back from the mountain region to the basin (Bei et al., 2017).
Leung et al. (2018) also find strong correlations of daily $PM_{2.5}$ variability with several
synoptic patterns, including monsoon flows and cold front channels in northern China
related to the Siberian High, onshore flows in eastern China, and frontal rainstorms in
southern China. These previous studies have highlighted that different levels of $PM_{2.5}$
pollutions are closely related to the dominant synoptic patterns in different regions, and
they attribute the large spatial variability of pollution to the regional transport
contributions, not only the different local sources of $PM_{2.5}$. Thus, heavy pollution
prevention and control needs to consider the weather situation, otherwise local emission
reduction measures would not work well. However, under different synoptic
conditions, how to effectively reduce local and regional emissions to control haze
pollution is rarely reported.
Various key regions have issued the emergency preplan against the winter haze
episodes, while these schemes can only be targeted at a certain city (The People's
Government of Beijing Municipality, 2018; The People's Government of Shanghai
Municipality, 2018) or a certain urban agglomeration (The People's Government of
Guangdong Province, 2014). Although there are many studies targeted $PM_{2.5}$
mitigations at a regional scale (Ding et al., 2019; Zhang et al., 2019, Xing et al., 2018,
2019; Fu et al., 2017; etc.), their results can not be directly applied to reduce winter
$PM_{2.5}$ pollution under various synoptic controls. Moreover, current emission reduction
policies in China mainly aimed at sulfur dioxide ($SO_2$) and nitrogen dioxide ($NO_2$),
ignoring the effective emission reduction on ammonia ($NH_3$), although some modeling
works have discussed the effectiveness of ammonia emission reduction for $PM_{2.5}$
mitigations (Liu et al., 2019; Ye et al., 2019; Xu et al., 2019; Bai et al., 2019). Compared
to remarkable reduction in $SO_2$, $NO_2$, and primary PM emissions, $NH_3$ emissions has
remained stable during 2014–2018 in China (Zheng et al., 2018). In addition, given the
important roles of other $PM_{2.5}$ precursors (eg., NMVOCs) in aerosol formation (Geng
et al., 2019), cutting non-$SO_2$-$NO_2$-PM emissions should be proposed as a next-step
mitigation strategy. Therefore, for $PM_{2.5}$ mitigations in a specific region during winter
haze episodes forced by various synoptic conditions, whether the air pollution
emergency management and control schemes are effective and how to improve them
have become an urgent scientific question to be answered.
In order to investigate the effectiveness of emission control to reduce $PM_{2.5}$
pollution during winter haze episodes under various potential synoptic controls (PSCs),
we take the severe particle pollution of winter haze episodes over Jingzhou, the
hinterland of Yangtze River middle basin in Central China, as an example. Central
China is geographically surrounded by major haze pollution regions, the SCB to the
west, the PRD to the south, the YRD to the east, and the NCP to the north (Fig. 1). As
a regional pollutant transport hub with sub basin topography, Central China is a region
of transmission-pollution characteristics affected by two reported transport pathways
from the vast flatland in central eastern China (Yu et al., 2020) and from the NCP region
(Zheng et al., 2019a). In combination with high anthropogenic emissions (Wu et al.,
2018) and secondary aerosol formation (Huang et al., 2020), Central China often suffers
severe pollution episodes in winter caused by $PM_{2.5}$ (Gong et al., 2015; Xu et al., 2017).
In this study, we conduct the circulation classification to differentiate the synoptic
modes during the severe particle pollution episodes in winter over Central China from
2013 to 2018 by using Lamb-Jenkension method. Then we simulate the $PM_{2.5}$ chemical
components, and the contributions of local sources as well as transboundary transport
of $PM_{2.5}$ under different synoptic conditions. Finally, the effectiveness of emission
reduction in main potential synoptic patterns are evaluated by GEOS-Chem model
simulations. This study combines the atmospheric (circulation classification) and
environmental (chemical transport modeling) research methods and could provide
reference for emission control of severe winter haze pollution under different weather
types, and provide basis for regional air quality policy-making.

**2 Data and Methods**

**2.1 Data**

Hourly mass concentrations of $PM_{2.5}$ at Jingzhou (112.18°E, 30.33°N, 33.7 m) from November 2013 to December 2018 are obtained from Hubei Environmental Monitoring Central Station (http://sthjt.hubei.gov.cn/). We screen the pollution days with daily mean $PM_{2.5}$ concentrations larger than 150 μg/m$^3$ for circulation classification.

Figure 1

We use the daily mean sea level pressure (SLP) between 2013 and 2018 from the National Centers for Environmental Prediction/National Center for Atmospheric Research (NCEP/NCAR) Final (FNL) Operational Global Analysis data (horizontal resolution: 1° × 1°; temporal resolution: 6 hours; https://rda.ucar.edu/datasets/ds083.3/) to conduct the classification of Lamb-Jenkension circulation types.

The meteorological data of surface observations at Jingzhou, including ambient temperature, relative humidity, wind speed, wind direction and atmospheric pressure, are obtained from Hubei Meteorological Information and Technology Support Center (http://hb.cma.gov.cn/qxfw/index.html). The data from November 2013 to February 2014 are used to analyze the meteorological characteristics during the period four severe particle pollution events occurred in succession over Central China (Fig. S1).

In order to better evaluate the GEOS-Chem model performances, we also use the $PM_{2.5}$ observations (a total of 633 sites; from November 2013 to February 2014) from Ministry of Ecology and Environment of China (MEE, http://www.mee.gov.cn/) to conduct the model-observation comparison.

**2.2 Lamb-Jenkension Circulation Classification**

170   The atmospheric circulation classification adopts the Lamb-Jenkension method

171 proposed by Lamb et al. (1950) and developed by Jenkension et al. (1977). Compared

172 to the objective classification method PCA used in some studies (Chang and Zhan, 2017,

173 Dai et al., 2021), this Lamb-Jenkension method is a combination of subjective and

174 objective methods. After the objective judgment of the circulation, we also make

175 subjective considerations to overcome the weaknesses of their respective, leading to

176 better synoptic significance. Many works of circulation classification have used the

177 Lamb-Jenkension method and reported that the analysis can well respond to the

178 classification results (Philipp et al., 2016;Santurtun et al., 2015;Pope et al., 2015;Russo

179 et al., 2014;Pope et al., 2014;Trigo and DaCamara, 2000).

180   To calculate the circulation types of Jingzhou, we mark total 16 points (97.5°E-

181 127.5°E, 20°N-40°N) by every 10 longitudes and 5 latitudes and the center point

182 located at 112.5° E and 30° N (Fig. S2). Using the sea level pressure of 16 points, we

183 calculate six circulation indexes by scheme of central difference:

184 $u = 0.5[P(12) + P(13) - P(4) - P(5)]$

185 (1)

186 $v = \frac{1}{\cos a} \times \frac{1}{4}[P(4) + 2P(9) + P(13) - P(4) - 2P(8) - P(12)]$

187 (2)

188 $V = \sqrt{u^2 + v^2}$

189 (3)

190 $\xi_u = \frac{\sin \alpha}{2\sin \alpha_1}[P(15) + P(16) - P(8) - P(9)] - \frac{\sin \alpha}{2\sin \alpha_2}[P(8) + P(9) - P(1) - P(2)]$

191 (4)

192 $\xi_v = \frac{1}{8\cos^2 \alpha}[P(6) + 2P(10) + P(14) - P(5) - 2P(9) - P(13)$
$+ P(3) + 2P(7) + P(11) - P(4) - 2P(8) - P(12)]$

193 (5)

$\xi = \xi_u + \xi_v$

(6)

Where $P(n)(n = 1, 2, 3 \cdots 16)$ is the sea level pressure at the $n^{th}$ point;
$\alpha, \alpha_1$ and $\alpha_2$ are the latitude values of points $C, A_1$ and $A_2$, respectively; $V$ is the
geostrophic wind, $u$ and $v$ are the latitudinal and meridional components of the
geostrophic wind; $\xi$ is the geostrophic vorticity; $\xi_u$ is the $u$ meridional gradient,
and $\xi_v$ is the $v$ latitudinal gradient.
Taking the latitude of the center point as the reference frame, the unit of six
circulation indexes is $hPa/(10^o lon)$, the direction of geostrophic wind can be
determined by $u$ and $v$, and cyclones and anticyclones can be determined by $\xi$.
According to the geostrophic wind speed, wind direction and vorticity value, the
circulation is divided into 10 types. The classification standard and corresponding types
are shown in Table 1.

Table 1


**2.3 GEOS-Chem simulations**
The GEOS-Chem chemistry transport model is used
(http://acmg.seas.harvard.edu/geos/) to simulate the spatiotemporal distribution of
PM$_{2.5}$. The nested model, covering China (70°E-140°E, 15°S-55°N), is run with a
horizontal resolution of 0.25° latitude × 0.3125° longitude and 72 vertical layers. The
boundary condition of nested model is provided by the GEOS-Chem global model with
a horizontal resolution of 2° latitude × 2.5° longitude (Fig. S3). Both global and nested
simulations, driven by the GEOS-FP assimilated meteorological data, include detailed
tropospheric Ozone-NO$_x$-VOCs-HO$_x$-aerosol chemistry. More details are shown in
Yan et al. (2019). In the models, anthropogenic and natural sources are fully considered
in GEOS-Chem. Table S1 and Table S2 show a list of emission inventories in the global
model and nested simulation, respectively. In China, the monthly grid data of $0.25° \times$
$0.25°$ from MEIC inventory (http://meicmodel.org) for CO, $NO_x$, $SO_2$ and non-methane
volatile organic compounds (NMVOCs) in 2013-2014 is used. Over Central China,
anthropogenic sources of these species are from our group SEEA (Source Emission and
Environment Research) inventory with the grid data of $0.1° \times 0.1°$ (not shown). The
SEEA emission inventory was developed based on the year of 2017 for the Wuhan city
cluster and it has been successfully adopted for the air quality simulating and
forecasting of 7th CISM Military World Games in 2019. Other emission descriptions
are shown in Supplementary Sect. S1.
In order to better simulate the spatiotemporal distribution of $PM_{2.5}$ over Central
China, especially in winter heavy pollution periods, the standard v11-01 of GEOS-
Chem is optimized according to the actual situation in China (see details in
Supplementary Sect. S2), including optimizing $PM_{2.5}$ sources and increasing the
proportion of sulfate primary emission (Yan et al., 2020). The $PM_{2.5}$ primary
anthropogenic emissions enhance the $PM_{2.5}$ concentrations over Central China by 5-20
$\mu g/m^3$ in winter (Fig. S4). Compared with the results before the model optimization
(Fig. S5), the sulfate concentration simulated by the optimized model increased from
10-20 $\mu g/m^3$ to 30-50 $\mu g/m^3$. Further comparisons of $PM_{2.5}$ with observations and
inorganic salts (sulfate, nitrate and ammonium) with reported values from previous
studies are shown in Sect.3.3.

**3. Results and Discussion**
**3.1 Classification of PSCs**
As shown in Fig. 2, among the circulation patterns of pollution-day at Jingzhou
from 2013 to 2018, the frequency of SW-type circulation is the highest, accounting for
29% of the total pollution days. The frequencies of NW-type, A-type and C-type are
also high, accounting for 27%, 19% and 12%, respectively. While the other six
circulation patterns are less occurred, with the frequencies less than 5%. Thus, the
above four typical circulation types are considered as the main potential synoptic
controls of the severe particle pollution episodes over Central China.

Figure 2


**3.2 Characteristics of the four main PSCs**

SW-type circulation is the predominant PSC of severe $PM_{2.5}$ pollution episodes.

The circulation at 500 hPa is relatively flat and the whole East Asia region is affected
by the westerly flow (Fig. S6a). Westerly belt fluctuates greatly at 700 hPa and there
are two ridges and a southwest trough in the middle latitudes of Asia (Fig. S7a).
Jingzhou is located in the front of a trough, prevailing the weak southwest airflow. At
850 hPa, the cold high pressure center is formed in Xinjiang of China. Warm low
pressure appears in the low latitude area and weak high pressure appears in the East
China Sea (Fig. 3a). In combination with the surface field, a high-low-high saddle like
field forms from west to east (Fig. 4a). Such synoptic type is also the dominant weather
system of eastern China (Shu et al., 2017; Yang et al., 2018). Jingzhou is located in the
back of Bohai-northeast high pressure and the front of southwest warm low pressure.
Thus it is affected by the southerly airflow, which could be conducive to the transport
of air pollutants formed over southern China to Central China. Associated with small
local surface wind speed (< 3 m/s) at Jingzhou, the dispersion of local and transported
pollutants is inhibited.

Figure 3


NW-type circulation mainly occurs in the early winter (December and January).

This synoptic pattern is also reported as one of the main types to affect the aerosol
distributions over eastern China (Zheng et al., 2015). Circulation at 500 hPa is
controlled by one trough and one ridge, with the weak ridge located in the northwest of
China and the shallow trough located in the northeast of China (Fig. S6b). The whole
East Asia is affected by the westerly current. The trough and ridge at 700 hPa are
deepened. Jingzhou is located at the bottom of the shallow trough, prevailing the west-
northwest airflow, affected by the flow around the plateau (Fig. S7b). At 850 hPa, the
cold high pressure center is formed in Xinjiang, and Jingzhou is affected by the
northerly airflow, due to being in the front of the high pressure (Fig. 3b). For the sea
level pressure, the cold high pressure is located in the west of Mongolia and Xinjiang
of China (Fig. 4b). Jingzhou is located at the region with weak fluctuation in the front
of the high pressure, and the surface wind speed is smaller than 2 m/s. The haze episodes
induced by NW-type synoptic pattern is similar to the transmission-accumulation
pollution caused by SW-type, but the transmission path is from Northern China to
Central China.

Figure 4


A-type circulation also mainly occurs in the early winter. The high-altitude
circulation field is controlled by one trough and one ridge (Fig. S6c and S7c). East Asia
is affected by west-northwest air flow, and the SLP is controlled by a huge high
pressure, with the center located in the southwest of Baikal Lake (Fig. 4c). A surface
high pressure favors accumulation of air pollutants, especially over the regions of high
pressure center (Leung et al., 2018). Jingzhou is in the sparse pressure field in front of
the high pressure (Fig. 3c and 4c), with an average surface wind speed of ~1.3 m/s. The
uniform west-northwest air flow at high altitude would lead to the low water vapor
content and less cloud amount, which is conducive to radiation cooling at night. In
addition, due to the weak high pressure ridge in the north, it is not conducive to the
eastward and southward movement of cold air, leading to the stable weather situation
and thus severe haze pollution at Jingzhou. This type is also responsible for most of the
severe particulate pollution days in the BTH and YRD regions (Li et al., 2019).

C-type circulation mainly occurs in late winter and early spring, when the relative

humidity is large with an average value of 74%. East Asia is controlled by the straight
westerly flow, and the southwest shallow trough is obvious at 500 hPa (Fig. S6d).
Additionally, the West Pacific subtropical high extends to the west, Central China is
affected by the southwest flow. Southwest trough is deepened at 700 hPa, and Jingzhou
is located in front of the trough and controlled by the southwest airflow (Fig. S7d). High
pressure at the south of Xinjiang and the north of Plateau is strengthened at 850 hPa,
and the southwest low pressure center is formed (Fig. 3d). Jingzhou is located in the
low pressure system on the SLP field (Fig. 4d), with small surface wind speed (0-3 m/s).
Together with the large relative humidity, which can promote the hygroscopic growth
of particulate matter (Twohy et al., 2009; Zheng et al., 2015), the haze pollution is
persistent and serious at Jingzhou. The impact of low-pressure systems on winter heavy
air pollution have also been reported in the northwest Sichuan Basin (Ning et al., 2018).

**3.3 $PM_{2.5}$ and chemical components under the four main PSCs in control**
**simulations**

The spatiotemporal distribution of $PM_{2.5}$ and its components under the four typical

synoptic controls over Central China were simulated by optimized GEOS-Chem model.
In order to reduce the simulation cost, the continuous four severe haze episodes
occurred during November, 2013-February, 2014 are selected. These four haze
episodes are controlled by the synoptic pattern of SW-type (18-25 November, 2013),
NW-type (19-26 December, 2013), A-type (14-21 January, 2014) and C-type (26
January - 2 February, 2014), respectively. The air quality at Jingzhou during the four
pollution episodes is between grade 5 ($PM_{2.5} > 150$ μg/m$^3$) and grade 6 heavy pollution
($PM_{2.5} > 250$ μg/m$^3$, as Fig. 5a and S1a shown). The simulation time is started at
November 1st, 2013, with the first two weeks used as spin up to eliminate the impact
of initial conditions.

Figure 5

Figure 6


The daily/hourly mean $PM_{2.5}$ concentrations at Jingzhou in the four typical heavy
pollution processes simulated by the control (CON) simulation (Table 2) are compared
with the observations (Fig. 5a/Fig. S1a). The model underestimates the observed $PM_{2.5}$
concentrations (by 43.3 μg/m$^3$ on average), especially in the high $PM_{2.5}$ periods (by
116.8 μg/m$^3$ at the maximum occurring in November 21-23, 2013). The possible causes
for underestimation are meteorological field deviations (an overestimate in temperature
and wind speed and an underestimate in humidity; Table S3) and emission errors.
Anthropogenic emissions for $PM_{2.5}$ precursors used here are for the year 2017 over
Central China from SEEA inventory (Table S4). From 2013 to 2017, anthropogenic
$NO_x$, $SO_2$, and primary $PM_{2.5}$ emissions in Central China have declined substantially
(Table S4), due to implementation of stringent emission control measures for the 12[th]-
13[th] Five-Year Plans (Zheng et al., 2018). The anthropogenic emissions biases may
affect our simulations and $PM_{2.5}$ attribution results to some extent. Aditionally, the
underestimation is on a national scale when compared with the MEE observations, with
a bias of -29.3 μg/m$^3$, -18.7 μg/m$^3$, -39.0 μg/m$^3$ and -21.4 μg/m$^3$ on average for SW-
type, NW-type, A-type and C-type synoptic controlled episodes, respectively (Fig. 6).
The national negative biases may be also attributed to insufficient resolution of the
model (Yan et al., 2014) and imperfect chemical mechanisms (Yan et al., 2019).
Nevertheless, the model can reproduce the evolution of each severe particle pollution
episode well, including the accumulation of pollutants, the continuing process and the
gradual dissipation of pollution (Fig. 5a/Fig. S1a).

Table 2


In order to examine the model performances in the $PM_{2.5}$ chemical compositions,

we have reviewed the reported concentrations of $PM_{2.5}$ and the three inorganic salts
(sulfate, nitrate and ammonium) in other cities (Table 4). The contributions of sulfate,
nitrate and ammonium are 9.1%-31.9%, 5.7%-32.1% and 5.9%-13.3%, respectively.
Figure S8/S9 shows the modeled spatial distribution of $PM_{2.5}$, sulfate, nitrate and
ammonium concentrations averaged in the four typical heavy pollution processes over
Jingzhou/China. The fractions of each inorganic salt to $PM_{2.5}$ for these four heavy
pollution episodes are also shown in Fig. S10. Over Central China, the main
components of $PM_{2.5}$ are the three inorganic salts in these pollution episodes, with the
averaged contributions of sulfate, nitrate and ammonium being ~20%, ~18% and ~13%,
respectively (Table 3). Our modelling results are comparable to the previous observed
results (Table 4). Huang et al. (2014) have also reported that the three secondary
inorganic particles rank the highest fraction among the $PM_{2.5}$ species in Central-Eastern
China. As shown in Table 3, in addition to inorganic salts, other chemical components
include dust (~15%), black carbon (~7%), primary organic aerosol (~14%) and second
organic aerosol (~13%). In these four pollution events, the differences in mass
percentages of each chemical component ranged from 0.1% (dust) to 6.2% (sulfate)
(Table 3). See details in Sect. 3.4 for further analysis of the causes for the differences.

Table 3

Table 4


**3.4 Local emissions versus transmission contributions to $PM_{2.5}$ under the four**
**main PSCs**

In order to investigate the effectiveness of emission control to reduce $PM_{2.5}$

pollution of Central China in the four typical severe particle pollution episodes, firstly
we estimate the local sources versus transmission contributions of $PM_{2.5}$ by GEOS-
Chem sensitivity simulations (Table 2). Results of XJ0 (Emissions outside Jingzhou are
zero) indicates the contribution of local emission sources to the $PM_{2.5}$ pollution over
Jingzhou. The difference between CON and XCC0 (Emissions outside Central China
are zero) shows the transmission contribution of $PM_{2.5}$ outside Central China to
Jingzhou. The difference between CON and NCP0/YRD0/PRD0/SCB0 (Emissions
over NCP/YRD/PRD/SCB are zero) represents the contribution of pollution transport
from NCP/YRD/PRD/SCB regions to Jingzhou.

Figure 7


For the SW-type synoptic situation, differences between the simulation results of

NCP0/YRD0/SCB0 and CON show that pollution controlled by SW-type circulation
over Central China is almost not affected by the emission sources from North
China/East China/Sichuan Basin. The concentrations of $PM_{2.5}$ and three inorganic salts
simulated by NCP0/YRD0/SCB0 are similar to those simulated by CON, with a
difference less than 3.0% (Fig. 8). However, affected by the southerly airflow at 850
hPa (Fig. 7), air pollutants formed over southern China could be transmitted to Central
China, with the transport contribution of 7.6%. In addition, the contributions from
transboundary transport from non-Jingzhou Central China is simulated to be 12.0% by
comparing the results of XJ0 and XCC0. The transport of air pollutants from the south
leads to the smallest proportion of the three inorganic salts (45.7%) in Jingzhou among
the four pollution episodes (50.3%-55.5% for other three episodes), because the
emissions of $SO_2$, $NO_2$ and $NH_3$ in the south (especially in Guangxi and Guizhou
province) are smaller than those in Central China (Li et al., 2017a). Associated with the
small surface wind speed of 2.1 m/s on average (Fig. 5) and the weak ascending in the
vertical direction (Fig. 7) at Jingzhou, it is not conducive to the dispersion of local
pollutants (Zheng et al., 2015). The high $PM_{2.5}$ concentrations are mainly accumulated
by local emissions. The simulations of XJ0 and CON show that local emission sources
over Jingzhou contribute ~70% to $PM_{2.5}$.

Figure 8

Figure 9


For the NW-type synoptic mode, affected by the northerly airflow (Fig. 9), it is

conducive to the southward movement of air pollutants in northern China ( He et al.,
2018; Leung et al., 2018). Influenced by the local and surrounding terrain over Central
China (Fig. 1), two transmission channels are formed from north to south and from
northeast to southwest (Fig. 9). In addition, due to the local small wind speed (1.4 m/s
on average) near the ground (Fig. 5), the weak convection and the warm ridge along
the East Asia coast (Fig. 9), the local and transported pollutants accumulate in Central
China. The average concentration of $PM_{2.5}$ in Jingzhou is 179.4 $\mu g/m^3$. Due to the
transport contribution of pollutants from northern China (with much higher
anthropogenic emissions of $SO_2$, $NO_2$ and $NH_3$) (Li et al., 2017a), the total proportion
of the three inorganic salts is the highest (55.5%). The $PM_{2.5}$ concentration simulated
in NCP0 is 63.1% of that by CON simulation (Fig. 8), indicating that the transmission
contribution from North China in this heavy pollution episode is as high as 36.9%. The
contribution of local emission sources is much smaller than that of SW-type synoptic
pattern, only 41.2% (comparison between XJ0 and CON).

Figure 10


Under the A-type circulation, Jingzhou is controlled by a high pressure system

(Fig. 10) which can lead to stable weather conditions caused by radiation inversion
(Guo et al., 2015) and subsidence inversion (Kurita et al., 1985), being favorable to
continuous accumulation of local pollutants (Guo et al., 2015). The distribution of $PM_{2.5}$
in China is similar to that of SW-type weather condition, with an averaged $PM_{2.5}$
concentration of 128.6 μg/m$^3$ over Central China. Unlike SW-type, the $PM_{2.5}$ at
Jingzhou in this synoptic pattern is less affected by transboundary transport, with the
total transport contribution of the surrounding four major pollution regions being less
than 9%. The contribution of local emission sources is about 82% (Fig. 8).

Figure 11


Under the C-type synoptic pattern, the southwest low pressure center is formed at
850 hPa, and Jingzhou is located in the low pressure system of the SLP field (Fig. 11).
In combination with the large relative humidity (78% on average; Fig. 5; because that
the occurrence season of C-type is the late winter and early spring), it can promote the
haze pollution due to its impact on hydrophilic aerosols (Twohy et al., 2009; Zheng et
al., 2015). Together with the small wind speed (less than 4 m/s; Fig. 5), it is easy to
cause the accumulation of pollutants. The average concentration of $PM_{2.5}$ over Central
China is as high as 203.7 μg/m$^3$. Air pollution controlled by this weather condition is
the most serious of the four typical synoptic controls. However, in this weather situation,
pollutants in North China are easy to diffuse (Miao et al., 2017; Li et al., 2019), and the
concentration of $PM_{2.5}$ is significantly lower than that in the former three weather
situations (Fig. 11 and Fig. S9). The contribution of pollution transport from non-
Central China region simulated by GEOS-Chem is less than 8%, and the contribution
of local emission sources at Jingzhou is more than 85% (Fig. 7).

**3.5 Effectiveness of emission reduction under the four main PSCs**
In order to estimate the effectiveness of emission reduction in severe pollution
events forced by the four potential synoptic controls, we conduct sensitivity simulations
by applying seven emission scenarios (Table 2). All emission scenarios use the
reduction ratio of 20% which is close to the average of the target emission reduction of
all provinces in the 13[th] Five-year plan (The State Council of the People's Republic of
China, 2016). Although the base year of emission reduction is 2015 for the 13[th] Five-
year plan, it does not affect to use the simulation results of emission scenarios (with the
reduction ratio of 20% applied to the simulated year 2013/2014) to explore the emission
reduction effect of specific haze pollution events. The differences in model results
between CON (control simulation) and JSN/JSNN/JALL (emissions of
$SO_2+NO_x$/$SO_2+NO_x+NH_3$/all pollution sources at Jingzhou are reduced by 20%)
represent the environmental benefits caused by different local emission reduction
scenarios. The potential $PM_{2.5}$ mitigations by joint prevention and control in different
regions are calculated by sensitivity experiments of CCALL (emissions of all pollution
sources over Central China are reduced by 20%), CNALL (over Central China and NCP
region), CPALL (over Central China and PRD region) and TALL (over Central China,
NCP, YRD, PRD and SCB regions).
In the JSN emission reduction scenario, the sulfate and ammonium concentrations
over Jingzhou are significantly reduced by 3.2-5.8 μg/m$^3$ (12.7-14.5%) and 0.6-1.9
μg/m$^3$ (3.2-5.9%) in these four pollution events, respectively. However, the
concentration of nitrate increases (1.3-1.7%). This is because there is a competition
mechanism between nitrate and sulfate. Ammonium ions always react with sulfate ions
first to generate ammonium sulfate, which will continue to react with nitrate ions to
generate ammonium nitrate when ammonium ions are rich (Mao et al., 2010). Thus the
reduction of $SO_2$ emission increases the concentration of nitrate, which offset the
contribution of sulfate particle reduction to the environment to some extent. Therefore,
the application of JSN emission reduction scheme only reduces the $PM_{2.5}$
concentrations by 3.1-7.2 μg/m$^3$ (2.0-3.5%, Fig. 12). This inefficient emission reduction
scheme is most widely used in heavy pollution areas over China in the past decade,
ignoring the synergistic effect of various precursors.

Figure 12


By applying the JSNN and JALL emission reduction scenarios, we aim to evaluate

the synergistic effect of multiple precursors on emission reduction. These two scenarios
reduce the average sulfate concentration in Jingzhou by 2.8-6.7 $\mu g/m^3$ (11.3-17.3%)
and 2.9-7.2 $\mu g/m^3$ (11.7-17.9%), and the ammonium concentration by 2.0-4.8 $\mu g/m^3$
(12.1-16.5%) and 2.2-4.7 $\mu g/m^3$ (13.2-17.3%), respectively. Unlike the increments of
nitrate in JSN emission reduction scenario, the nitrate decreases (JSNN: 0.3-1.2 $\mu g/m^3$;
JALL: 0.4-1.5 $\mu g/m^3$). Therefore, through the application of JSNN and JALL emission
reduction schemes, $PM_{2.5}$ concentrations decrease by 4.9-8.3% and 9.0-15.9%,
respectively (Fig. 12), much higher than the improvement by JSN scenario. Zheng et
al. (2019b) has also evaluated the sensitiveness of $NH_3$ control to $PM_{2.5}$ reduction based
on observations. However, these results indicate that it is unrealistic to substantially
reduce local emissions to achieve the national air quality standard in the long term.

Additionally, the sensitivity simulations by excluding emission sources over

upwind regions are conducted to estimate the potential $PM_{2.5}$ mitigations of inter-
regional and intra-regional joint control. Our results show that after applying TALL
emission reduction scenario, $PM_{2.5}$ concentrations have been significantly improved,
with the improvement rates increased from 9.0-15.9% (by JALL scenario) to 17.4-18.8%
(Fig. 12). Especially, the NW-type synoptic controlled air pollution episode shows the
best effect of joint prevention, followed by SW-type. For NW-type, by reducing
emissions over Central China and Northern China (CNALL scheme), $PM_{2.5}$
concentrations are reduced by 26.5 $\mu g/m^3$ (16.9%), much more effective than JALL
emission reduction scheme (14.1 $\mu g/m^3$, 9.0%). In SW-type controlled pollution
episode, it should be otherwise to decrease the emissions over Southern China in
addition to Central China.

**4. Conclusion**
The PM$_{2.5}$ pollution in autumn and winter haze periods is now the key obstacle for
further improving air quality in China. The extremely severe and persistent PM$_{2.5}$
pollution episodes are attributed to adverse synoptic conditions in addition to high
precursor emissions. For the PM$_{2.5}$ mitigations during winter haze episodes in specific
region forced by various potential synoptic controls, how to effectively reduce
emissions has become an urgent scientific question to be answered. Our results over
Central China could provide reference for regional air quality policy-making.
Through Lamb-Jenkension circulation classification, the top four potential
synoptic controls of heavy PM$_{2.5}$ pollution days (totally 109 days) over Central China
from 2013 to 2018 are decomposed to be SW-type, NW-type A-type and C-type,
accounting for 29%, 27%, 19% and 12% of the total pollution days, respectively. In
these four PSCs, three inorganic salt aerosols (sulfate: ~20%; nitrate: ~18%;
ammonium: ~13%) totally accounted for ~51% of PM$_{2.5}$ concentrations simulated by
optimized GEOS-Chem modelling.
In the SW-type/NW-type synoptic situation, affected by the southerly/northerly
airflow, pollutants over southern/northern China could be transmitted to Central China,
with the transport contribution of 7.6%/37%. In the situation A-type/C-type weather,
affected by stable weather condition/high relative humidity, the pollution processes are
less affected by the emission sources from non-local regions. And the local emission
sources dominate the contribution (82%/85%) to PM$_{2.5}$.
By only reducing SO$_2$ and NO$_x$ emission and not controlling NH$_3$, due to the
competition mechanism between nitrate and sulfate, the concentrations of sulfate and
ammonium decrease, but the concentration of nitrate increases instead. The enhanced
nitrate counteracts the effect of sulfate reduction on PM$_{2.5}$ mitigations, with less than 4%
decrease in PM$_{2.5}$. Even if the NH$_3$ emission is also reduced, the PM$_{2.5}$ concentration
reduction is less than 9%. By applying the TALL emission reduction scenario, PM$_{2.5}$
concentrations would decrease significantly, with the improvement rate increased from
9.0-15.9% (by JALL scenario) to 17.4-18.8%.
These results provide an opportunity to effectively mitigate haze pollution by local
emission control actions in coordination with regional collaborative actions according
to different synoptic patterns. Especially, the NW-type synoptic controlled air pollution
episode shows the best effect of joint prevention, followed by SW-type. It is noted that
in this study, the division of transmission areas is relatively rough, and more accurate
source area identification and refined assessment of emission reduction effect of
multiple pollutants from source groups are needed in the follow-up.

**Acknowledgement**
This study was financially supported by the National Natural Science Foundation
of China (41830965; 41775115; 41905112), the Key Program of Ministry of Science
and Technology of the People's Republic of China (2017YFC0212602;
2016YFA0602002), the Key Program for Technical Innovation of Hubei Province
(2017ACA089), the Program for Environmental Protection in Hubei Province
(2017HB11) and the China Postdoctoral Science Foundation funded project (258572).
The research was also funded by the Fundamental Research Funds for the Central
Universities, China University of Geosciences (Wuhan) (G1323519230; 201616;
26420180020; CUG190609) and the Start-up Foundation for Advanced Talents
(162301182756).


**Author contributions**
Yingying Yan and Shaofei Kong conceived and designed the research. Yingying
Yan performed the data processing, model simulations, and analyses. Yue Zhou
assisted in the circulation classification. Jian Wu provided the emission data over
Central China. Shaofei Kong, Tianliang Zhao and Dantong Liu contributed the funding
acquisition. Yingying Yan wrote the paper with input from all authors.

**Data availability**
Observational data are obtained from individual sources (see links in the text).
Model results are available upon request. Model codes are available on a collaborative
basis.

**Competing interests**
The authors declare that they have no conflict of interest.

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

Figure 2 Frequency distributions of ten circulation types for the heavy pollution days
of 2013-2018 over Jingzhou. The occurrence numbers of each type are shown. The ten
circulation types include Southwest (SW), Northwest (NW), Anticyclone (A), Cyclone
(C), Anticyclone-West (AW), Cyclone-West (CW), Cyclone-Southeast (CSE),
Cyclone-Northwest (CNW), Southeast (SE) and East (E), respectively.

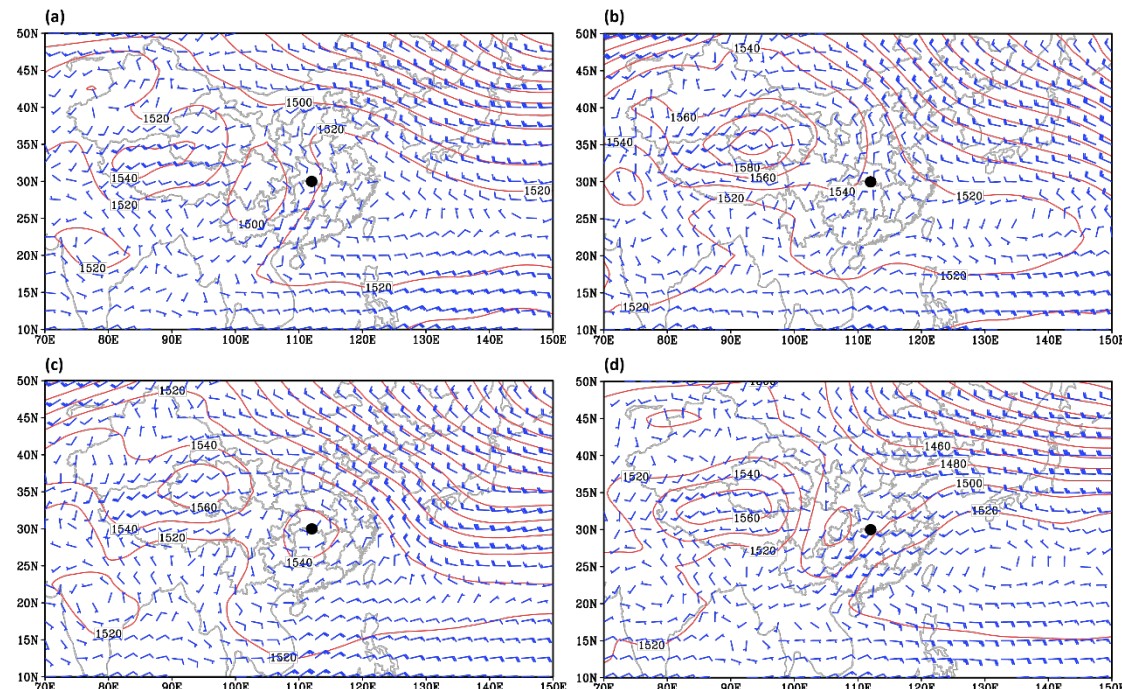

Figure 3 Spatial distribution of 850 hPa geopotential height and wind vector for SW-
type (a), NW-type (b), A-type (c) and C-type (d) synoptic control averaged over 2013-
2018. The black dot indicates the location of Jingzhou.

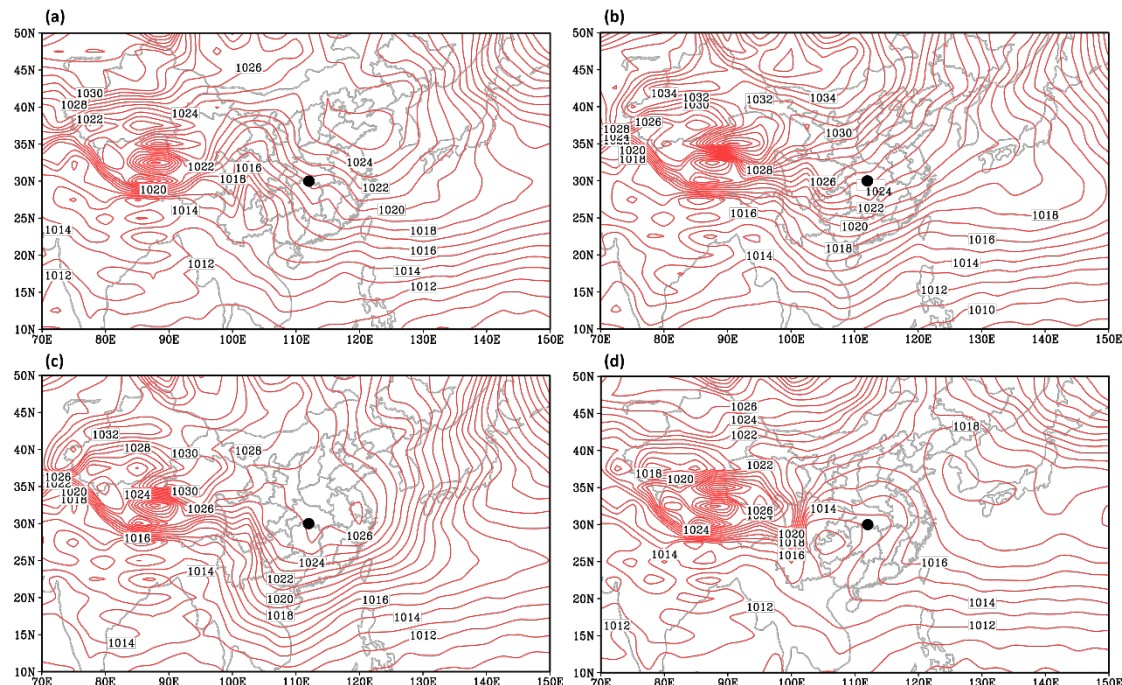

Figure 4 Spatial distribution of sea level pressure for SW-type (a), NW-type (b), A-type

(c) and C-type (d) synoptic control averaged over 2013-2018. The black dot indicates

the location of Jingzhou.

956

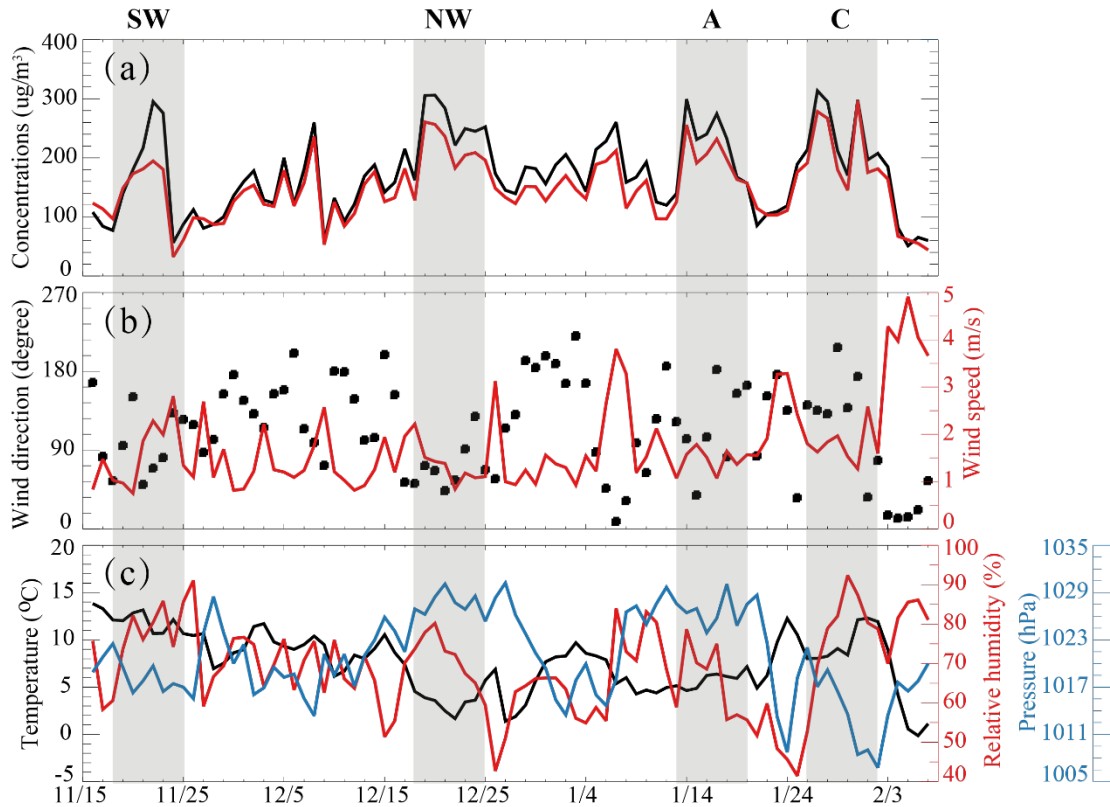

957

Figure 5 (a) Daily mean values of modeled (red line) and observed (black line) PM$_{2.5}$ concentration (μg/m$^3$) at Jingzhou and four severe pollution events (grey area) from November, 2013 to February, 2014. (b) Observed daily mean wind speed (red line) and wind direction (black dots). (c) Obseved temperature (black line), relative humidity (red line) and sea level pressure (blue line).



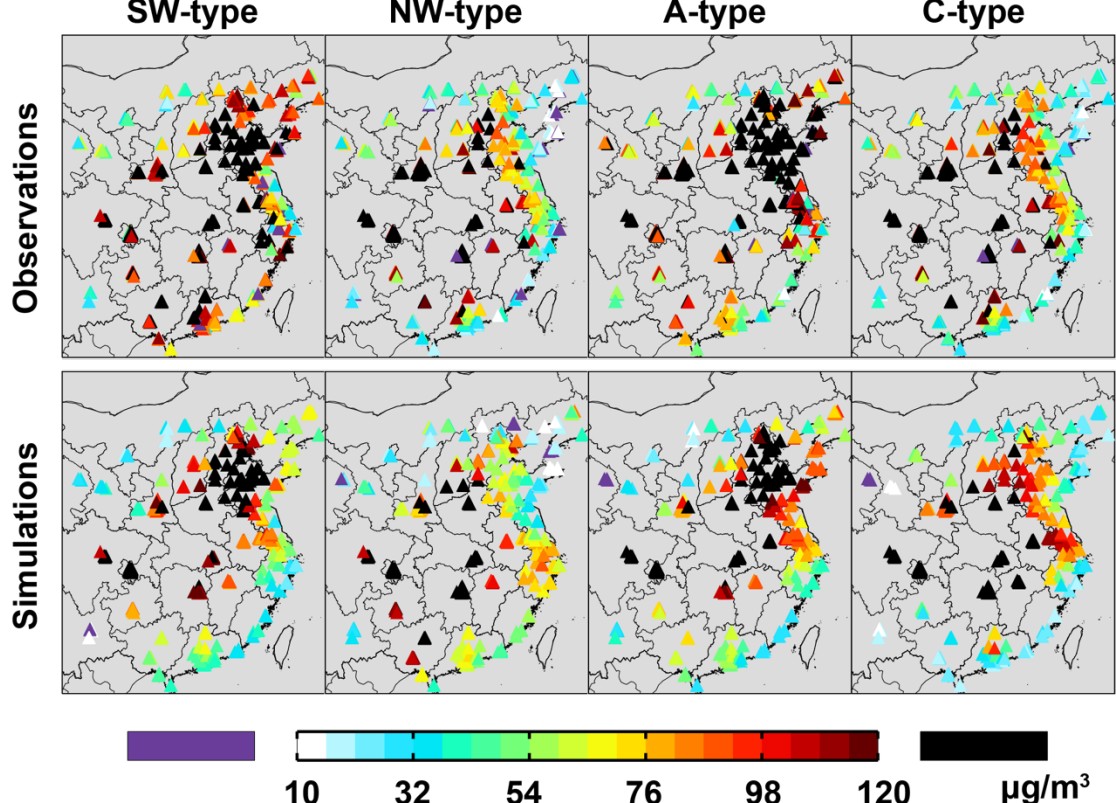

Figure 6 Spatial distribution of observed (top row) and modeled (bottom row, by CON
case) PM$_{2.5}$ concentrations (μg/m$^3$) averaged over four severe pollution episodes
controlled by SW-type (first column), NW-type (second column), A-type (third
column) and C-type (forth column) synoptic pattern, respectively.

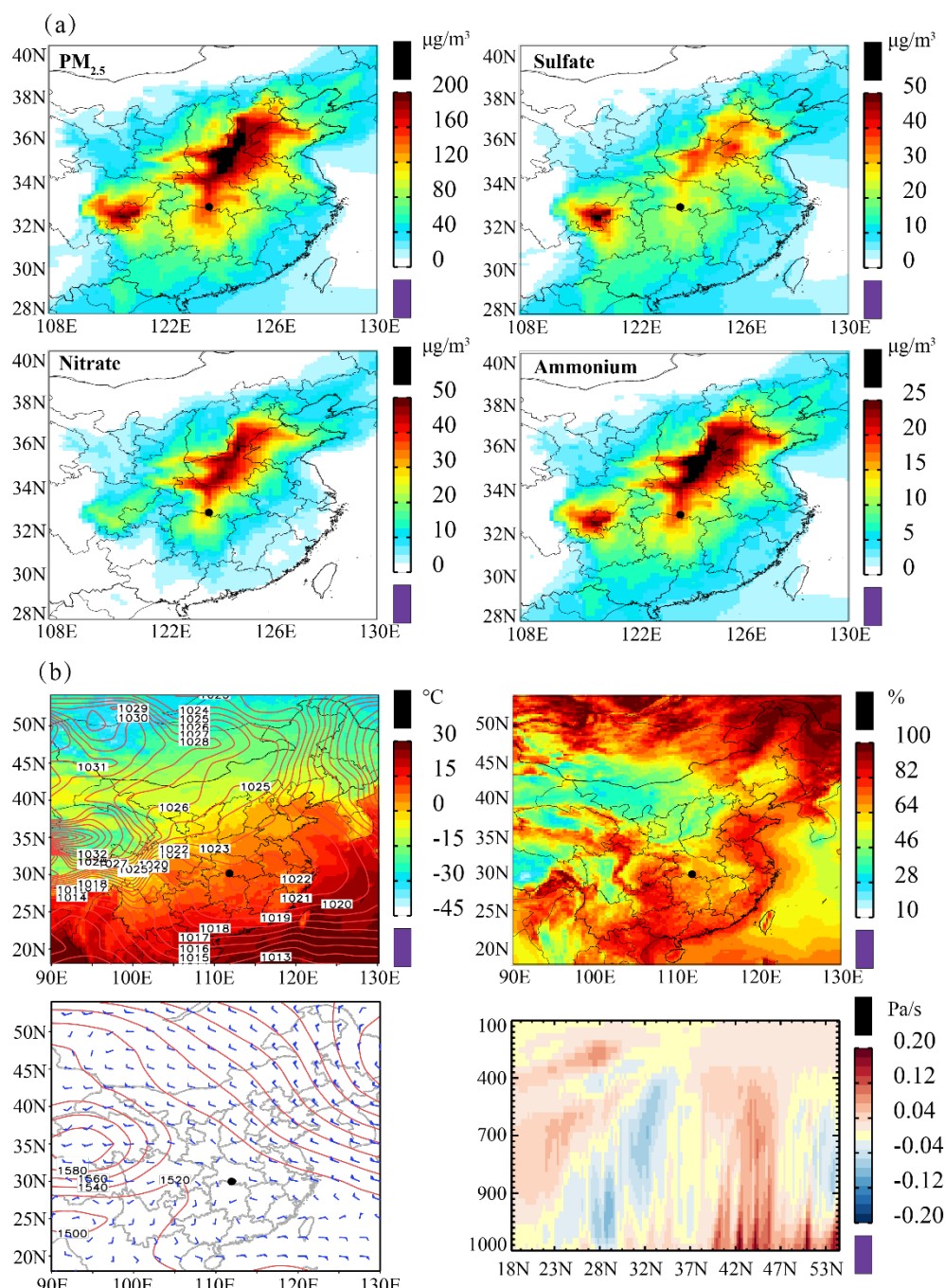

Figure 7 (a) Spatial distribution of $PM_{2.5}$, sulfate, nitrate and ammonium concentrations averaged over SW-type synoptic controls (18-25 November, 2013) simulated by GEOS-Chem control simulation ($\mu g/m^3$). (b) Meteorological conditions of SW-type: sea level pressure (red line) and temperature (colour shades), surface relative humidity (%) fields, 850 hPa wind and geopotential height (red line) and height–latitude cross-sections of vertical velocity (Pa/s).

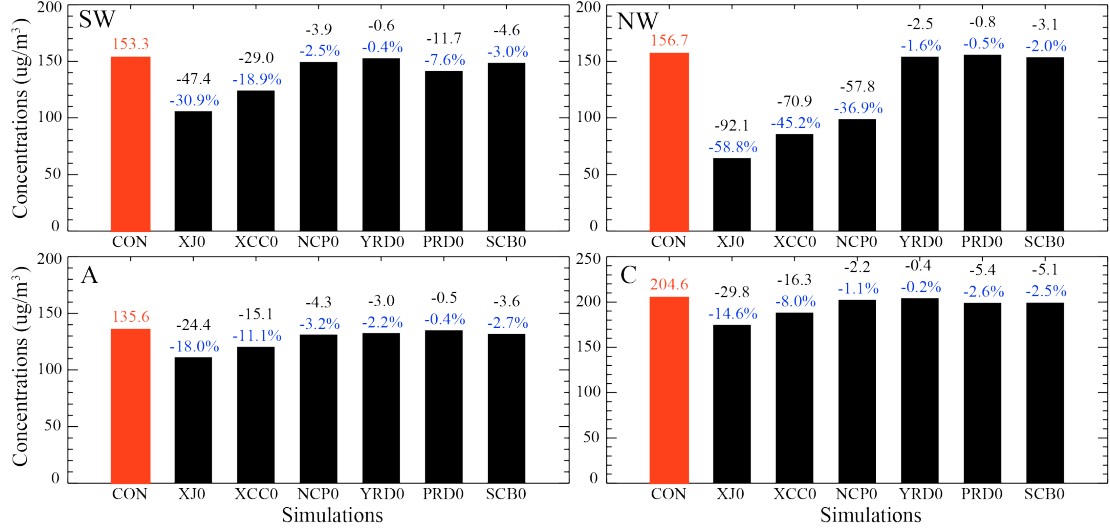

Figure 8 Modeled concentrations (μg/m³) of PM$_{2.5}$ at Jingzhou in the GEOS-Chem
control (red bar) and sensitivity (black bar) simulations in view of the regional
transportation, and the differences (black characters for mass concentrations and blue
characters for mass percentages) between the sensitivity and the control simulations.
The abbreviations of each simulation referred to Table 2.



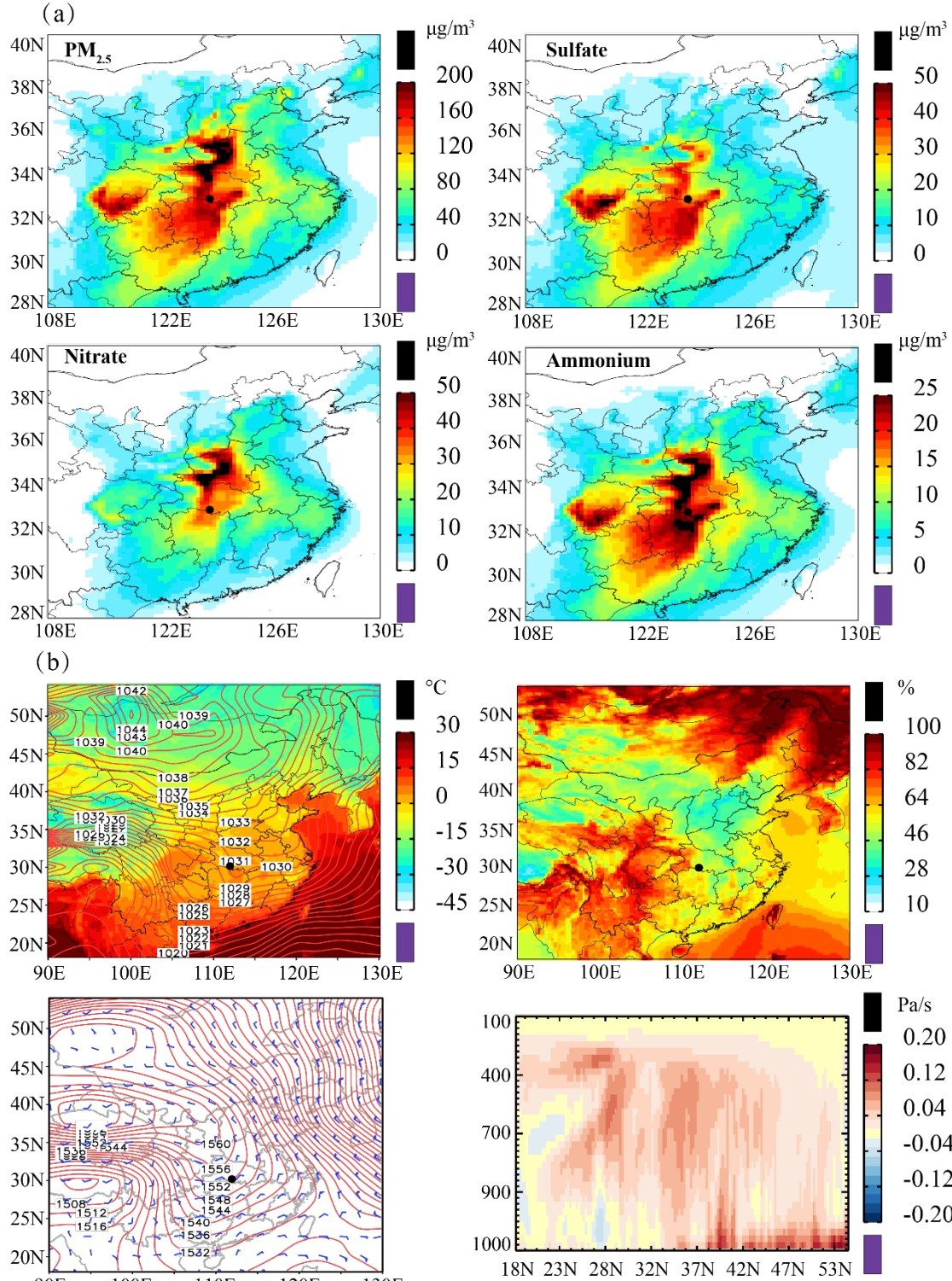

Figure 9 As in Fig. 6 but for NW-type synoptic control (19-26 December, 2013).



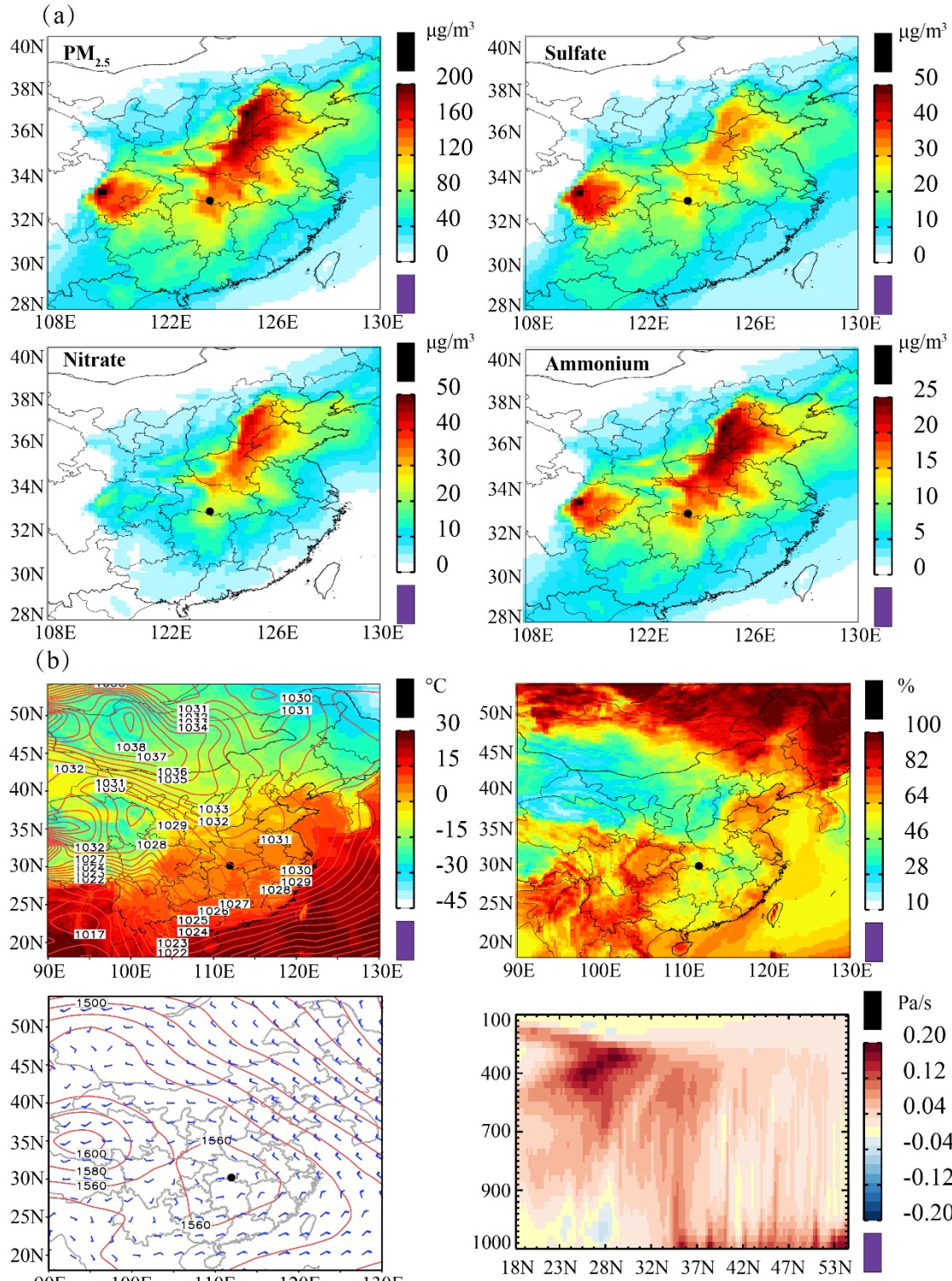


Figure 10 As in Fig. 6 but for A-type synoptic control (14-21 January, 2014).



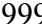

Figure 11 As in Fig. 6 but for C-type synoptic control (26 January - 2 February, 2014).


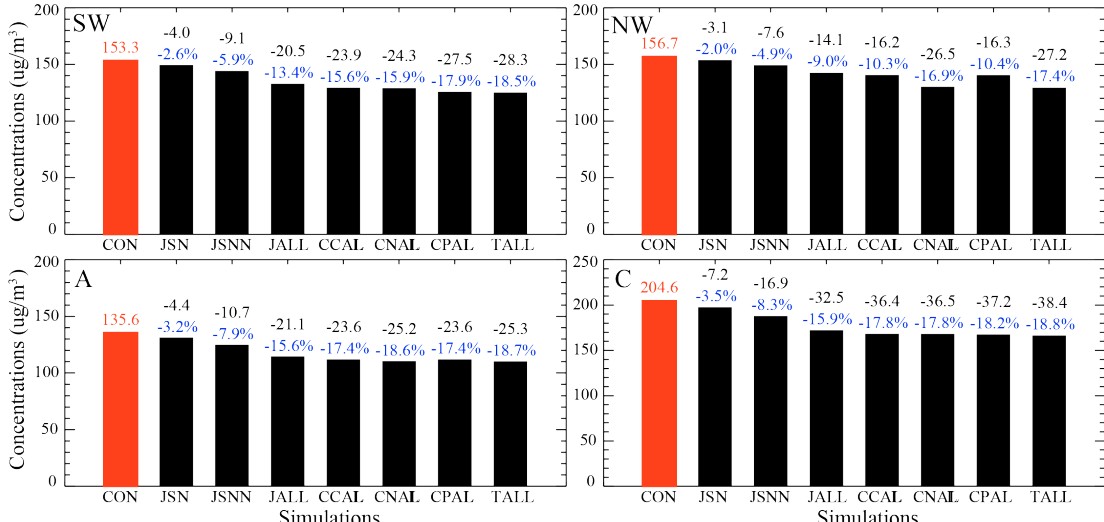

Figure 12 Modeled concentrations (μg/m³) of PM₂.₅ at Jingzhou in the GEOS-Chem
control (red bar) and sensitivity (black bar) simulations for emission reduction, and the
differences (black characters for mass concentrations and blue characters for mass
percentages) between the sensitivity and the control simulations. The abbreviations of
each simulation referred to Table 2.



1014             Table 1 Lamb-Jenkinson circulation types

| $\lvert \xi \rvert \leq V$ | $\lvert \xi \rvert \geq 2V$ | $V < \lvert \xi \rvert < 2V$ |
|---|---|---|
| (Flat airflow type) | (Rotating airflow type) | (Mixed type) |
| East (E), | Anticyclone (A), | Cyclone-Southeast (CSE), |
| Southeast (SE), | Cyclone (C) | Cyclone-West (CW), |
| Southwest (SW), | | Cyclone-Northwest (CNW), |
| Northwest (NW) | | Anticyclone-West (AW) |




Table 2 Description of sensitivity simulations by GEOS-Chem model. The NCP, YRD,
PRD and SCB are the areas framed in red showed by Fig. 1.

| Simulations | Description |
| --- | --- |
| CON | Applying the original emission situation in Table S1 and Table S2 |
| XJ0 | Emissions of all pollution sources[1] outside Jingzhou are set to be zero |
| XCC0 | Emissions of all pollution sources outside Central China are set to be zero |
| NCP0 | Emissions of all pollution sources over NCP region are set to be zero |
| YRD0 | Emissions of all pollution sources over YRD region are set to be zero |
| PRD0 | Emissions of all pollution sources over PRD region are set to be zero |
| SCB0 | Emissions of all pollution sources over SCB region are set to be zero |
| JSN | Emissions of $SO_2$ and $NO_x$ at Jingzhou are reduced by 20% |
| JSNN | Emissions of $SO_2$, $NO_x$ and $NH_3$ at Jingzhou are reduced by 20% |
| JALL | Emissions of all pollution sources at Jingzhou are reduced by 20% |
| CCALL | Emissions of all pollution sources over Central China are reduced by 20% |
| CNALL | Emissions of all pollution sources over Central China and NCP region are reduced by 20% |
| CPALL | Emissions of all pollution sources over Central China and PRD region are reduced by 20% |
| TALL | Emissions of all pollution sources over Central China, NCP, YRD, PRD and SCB region are reduced by 20% |

1.   All pollution sources include emissions of $SO_2$, $NO_x$, $NH_3$, CO, BC, OC and NMVOCs.




Table 3 Simulated PM$_{2.5}$ concentrations and associated chemical components averaged
for the four typical heavy pollution episodes at Jingzhou. Also shown in brackets are
the percentages of each component in PM$_{2.5}$.

| PM$_{2.5}$ components | Typical heavy pollution episodes | | | |
|---|---|---|---|---|
| µg/m$^3$ | 11/18-11/25(SW-type) | 12/19-12/26(NW-type) | 1/14-1/21(A-type) | 1/26-2/2(C-type) |
| Nitrate | 30.6 (20.0%) | 34.6 (22.1%) | 23.4 (17.3%) | 42.3 (20.7%) |
| Sulfate | 26.5 (13.4%) | 30.7 (19.6%) | 27.7 (20.4%) | 40.4 (19.7%) |
| Ammonium | 18.8 (12.3%) | 21.6 (13.8%) | 17.1 (12.6%) | 27.1 (13.2%) |
| Dust | 24.4 (15.9%) | 22.3 (14.2%) | 19.8 (14.6%) | 29.2 (14.3%) |
| BC | 10.5 (6.8%) | 9.6 (6.1%) | 9.5 (7.0%) | 13.8 (6.7%) |
| POA | 21.6 (14.1%) | 18.9 (12.1%) | 18.9 (13.9%) | 27.7 (13.5%) |
| SOA | 20.9 (13.6%) | 19.0 (12.1%) | 19.2 (14.2%) | 24.1 (11.8%) |
| PM$_{2.5}$ | 153.3 | 156.7 | 135.6 | 204.6 |



Table 4 The reported concentrations of PM$_{2.5}$ and the three inorganic salts (sulfate,
nitrate and ammonium, μg/m$^3$) in other cities.

| References | Site | Time | PM$_{2.5}$ | Sulfate | Nitrate | Ammonium |
|---|---|---|---|---|---|---|
| Cao et al., 2012 | Beijing | 01/03 | 115.6±46.6 | 20.0±4.2 (17.3%) | 13.1±4.5 (11.3%) | 9.4±4.1 (8.1%) |
| Cao et al., 2012 | Qingdao | 01/03 | 134.8±43.0 | 21.1±7.7 (15.7%) | 19.3±9.2 (14.3%) | 15.3±5.2 (11.4%) |
| Cao et al., 2012 | Tianjin | 01/03 | 203.1±76.2 | 32.5±15.1 (16.0%) | 25.2±10.3 (12.4%) | 22.2±9.8 (10.9%) |
| Cao et al., 2012 | Xi'an | 01/03 | 356.3±118.4 | 53.8±25.6 (15.1%) | 29.0±10.0 (8.1%) | 29.8±11.5 (8.4%) |
| Cao et al., 2012 | Chongqing | 01/03 | 316.6±101.2 | 60.9±19.6 (19.2%) | 18.1±6.4 (5.7%) | 28.8±8.9 (9.1%) |
| Cao et al., 2012 | Hangzhou | 01/03 | 177.3±59.5 | 33.4±16.7 (18.8%) | 25.7±14.8 (14.5%) | 19.1±10.7 (10.8%) |
| Cao et al., 2012 | Shanghai | 01/03 | 139.4±50.6 | 21.6±12.3 (15.5%) | 17.5±8.7 (12.6%) | 14.5±5.9 (10.4%) |
| Cao et al., 2012 | Wuhan | 01/03 | 172.3±67.0 | 31.4±15.6 (18.2%) | 22.2±10.7 (12.9%) | 18.4±10.2 (10.7%) |
| Zhang et al., 2011 | Xi'an | 03/06-03/07 | 194.1 | 35.6 (18.3%) | 16.4 (8.4%) | 11.4 (5.9%) |
| Huang et al., 2012 | Xi'an | 01/06-02/06 | 235.8±125.1 | 44.8±31.3 (19.0%) | 20.5±14.2 (8.7%) | 14.5±10.8 (6.1%) |
| Wang et al., 2020 | Jinan | 10/17 | 104±54 | 14.4±9.2 (13.8%) | 33.4±23.2 (32.1%) | 13.0±8.3 (12.5%) |
| Wang et al., 2020 | Shijiazhuang | 10/17 | 152±109 | 19.3±19.6 (12.7%) | 42.8±41.1 (28.2%) | 18.2±17.1 (12.0%) |
| Wang et al., 2020 | Wuhan | 12/17 | 117±33 | 13.6±3.2 | 26.6±11.1 | 13.1±3.8 |

| | | | | | | |
|---|---|---|---|---|---|---|
| | | | | (11.6%) | (22.7%) | (11.2%) |
| Wang et al., 2016a | Zhengzhou | 01/11-02/11 | 297±160 | 48±36 (16.2%) | 31±19 (10.4%) | 21±16 (7.1%) |
| Wang et al., 2016a | Zhengzhou | 01/12-02/12 | 234±125 | 23±10 (9.8%) | 22±9 (9.4%) | 16±5 (6.8%) |
| Wang et al., 2016a | Zhengzhou | 01/13-02/13 | 337±168 | 56±39 (16.6%) | 39±20 (11.6%) | 31±18 (9.2%) |
| Luo et al., 2018 | Zibo | 12/06-02/07 | 224.9±85.4 | 40.1±19.2 (17.9%) | 18.1±9.0 (8.1%) | 21.7±10.2 (9.7%) |
| Wang et al., 2016b | Shanghai | 12/11, 12/12, 12/13 | 73.9±57.5 | 12.2±9.2 (16.5%) | 14.6±12.2 (19.8%) | 8.2±6.7 (11.1%) |
| Xu et al., 2019 | Beijing | 02/17-03/17 | 180.5 | 20.1 (11.1%) | 45.6 (25.3%) | 22.5 (12.5%) |
| Xu et al., 2019 | Beijing | 05/17-09/17 | 186.7 | 20.2 (10.8%) | 32.4 (17.4%) | 17.1 (9.2%) |
| Xu et al., 2019 | Beijing | 10/17-11/17 | 167.5 | 17.9 (10.7%) | 44.5 (26.6%) | 20.9 (12.5%) |
| Zheng et al., 2016 | Beijing | 03/10-05/10 | 65.2±65.1 | 11.1±10.1 (17.0%) | 11.1±11.0 (17.0%) | 6.8±6.7 (10.4%) |
| Zheng et al., 2016 | Beijing | 07/09-08/09 | 88.9±39.1 | 23.0±13.9 (25.9%) | 16.2±11.8 (18.2%) | 11.8±6.8 (13.3%) |
| Zheng et al., 2016 | Beijing | 12/09-02/10 | 84.0±66.6 | 8.1±8.3 (9.1%) | 8.0±9.6 (9.0%) | 5.9±7.1 (6.6%) |
| Zheng et al., 2016 | Guangzhou | 11/10 | 73.3±16.5 | 16.6±4.0 (22.6%) | 5.7±3.8 (7.8%) | 6.2±2.0 (8.5%) |
| Zheng et al., 2016 | Shenzhen | 12/09 | 64.6±24.7 | 20.6±3.5 (31.9%) | 4.9±3.5 (7.6%) | 4.6±1.0 (7.1%) |
| Zheng et al., 2016 | Wuxi | 04/10-05/10 | 82.1±27.0 | 12.8±3.8 | 9.9±6.3 | 7.0±2.0 |

| | | | | (15.6%) | (12.1%) | (8.5%) |
|---|---|---|---|---|---|---|
| Zheng et al., 2016 | Jinhua | 10/11-11/11 | 81.9±26.2 | 18.3±6.7 (22.3%) | 12.6±7.0 (15.4%) | 10.4±4.1 (12.7%) |
| Liu et al., 2018 | Chongqing | 2012-2013 | 73.5±30.5 | 19.7±9.6 (26.8%) | 6.5±6.2 (8.8%) | 6.1±2.7 (8.3%) |
| Liu et al., 2018 | Shanghai | 2012-2013 | 68.4±20.3 | 13.6±6.4 (19.9%) | 11.9±5.0 (17.4%) | 5.8±2.1 (8.5%) |
| Liu et al., 2018 | Beijing | 2012-2013 | 71.7±36.0 | 11.9±8.2 (16.6%) | 9.3±7.5 (13.0%) | 5.3±2.7 (7.4%) |
