# Peer review of "Effectiveness of emission control to reduce PM2.5 pollution of Central China"

_Atmospheric Chemistry and Physics, 2020_

## Referee Comment (RC1) · Anonymous Referee #1 · 11 Dec 2020

This paper by Yan et al. investigated the characteristics of winter haze episodes in Jingzhou of Central China under typical potential synoptic controls (PSCs) during November 2013-February 2014. Furthermore, they examined the contributions of local and transport of pollutants from surrounding regions to $PM_{2.5}$ under different PSCs by applying the GEOS-Chem model with a high resolution. This work also studied the effectiveness of different emission control strategies in Jingzhou, Central China, and other surrounding regions under different PSCs, and highlights the importance of collaborative actions for $PM_{2.5}$ mitigation under server haze pollution. In general, the study is well organized and worthy of publication. However, I have some specific comments that I feel deserve attention.

Major comments

1. The writing should be improved before publication.

2. The configuration of the model is vague. How many nested domains were applied in each simulation? What is the geographic coverage of each domain and the corresponding resolution? What are the emission inventories for each domain? A figure showing each nested domain is also highly recommended.

3. The circulation classification is the basis of all the analysis. Why did you choose the Lamb-Jenkension method? What are the advantages of this method compared to the ones used in other studies such as Chang

and Zhan, 2017, Dai et al., 2021, etc.?

4. The validation of model performances is very weak. The bias of the modeled $PM_{2.5}$ in Jingzhou can be as high as more than 100 $\mu g/m^3$, what are the possible reasons? The authors simply claimed the uncertainties in emissions, meteorology, and chemistry might cause this discrepancy without any details. What are the amount of the $PM_{2.5}$ precursors emitted in this study and how are the values compared to the published literature? How about the meteorological parameters used by the model vs. observations? The authors claimed an improvement in sulfate by the increase in primarily emitted sulfate in the model, how is that compared with observations? They also analyzed the changes in the chemical composition of $PM_{2.5}$ under different typical PSCs without examination of the model performances in the base case.

Minor comments:

Line 101-103: There must be many studies targeted the mitigation of $PM_{2.5}$ at a regional scale (Ding et al., 2019; Zhang et al., 2019, Xing et al., 2018, 2019; Fu et al., 2017; etc.). Please rephrase this sentence.

Line 148-150: It is very confusing. The circulation classification is based on the meteorological data from November 2013 to February 2014, which is also the simulation episode. Why did you use the hourly $PM_{2.5}$ data from 2013-2018?

Line 195: Did you do nested runs or just one domain covering China? Please make this clear.

Line 205: The SEEA inventory was developed for the year 2017. Did you use it directly without projection to the simulation episode? If you adjusted this inventory, what are the factors applied for the $PM_{2.5}$ precursors and how did you obtain those data?

Line 215-217: Have you compared the modeled sulfate with observations, at least in Jingzhou? How about the model performances of the other components of $PM_{2.5}$?

Line 305: Again, I am confused about the emissions used in the CON case. You listed too many options for the anthropogenic source in Table S2. What inventories were EXACTLY selected for the CON case? Did you do a global/regional nested run? Please explain the choices of emissions in a separate column in the table.

Line 310: Please compare the meteorological field used in the model with observations to confirm that statement. Also, there are no perfect mechanisms, inventories, or parameterization of the model with no doubt. I suggest using "uncertainties".

Line 323-324: A comparison of the modeled fractions of the inorganic salts to observations, or reported values from other literature if no measurements are available.

Line 324: "As shown in Table 3, …."

Line 358: How was this calculated? Please explain it.

Line 415-417: How about the contributions of transported pollutants to the chemical composition of $PM_{2.5}$ under the four PSCs?

Line 424: The base year of emission reduction is 2015 for the 13[th] Five-year plan, which is quite different from your inventory. How effective is the designed reduction ratio of the anthropogenic emissions in this study?

Line 425 and 428-429: Please explain these abbreviations in the text as well.

Line 437: I think an evaluation of the model performance in ammonium and/or ammonia is desired to confirm that.

Figure 6, 8, 9, 10: I suggest to show the fraction of each inorganic salt to $PM_{2.5}$ rather than their total mass.

Figure 11. It should be "TALL" in NW and C.

References:

Wenyuan Chang and Jianqiong Zhan, The association of weather patterns with haze episodes: Recognition by PM2.5 oriented circulation classification applied in Xiamen, Southeastern China, Atmospheric Research, 197, 425-436, 2017.

Huibin Dai, Jia Zhu, Hong Liao, Jiandong Li, Muxue Liang, Yang Yang, Xu Yue, Co-occurrence of ozone and PM2.5 pollution in the Yangtze River Delta over 2013–2019: Spatiotemporal distribution and meteorological

conditions, Atmospheric Research, 249, 105363, 2021.

Qiang Zhang, Yixuan Zheng, Dan Tong, et al., Drivers of improved PM2.5 air quality in China from 2013 to 2017, Proceedings of the National Academy of Sciences, 116 (49) 24463-24469, 2019.

Aijun Ding, Xin Huang, Wei Nie, et al., Significant reduction of PM2.5 in eastern China due to regional-scale emission control: evidence from SORPES in 2011–2018, Atmospheric Chemistry and Physics, 19, 11791–11801, 2019.

Xing, J., Ding, D., Wang, S., Dong, Z., Kelly, J. T., Jang, C., ... & Hao, J. Development and application of observable response indicators for design of an effective ozone and fine-particle pollution control strategy in China. Atmospheric Chemistry & Physics, 19(21), 2019.

Xing, J., Ding, D., Wang, S., Zhao, B., Jang, C., Wu, W., ... & Hao, J. Quantification of the enhanced effectiveness of NO x control from simultaneous reductions of VOC and NH 3 for reducing air pollution in the Beijing–Tianjin–Hebei region, China. Atmospheric Chemistry and Physics, 18(11), 7799-7814, 2018.

Fu, X., Wang, S.,Xing, J., Zhang, X., Wang, T., & Hao, J. Increasing ammonia concentrations reduce the effectiveness of particle pollution control achieved via SO2 and NO X emissions reduction in east China. Environmental Science & Technology Letters, 4(6), 221-227, 2017.

---

## Referee Comment (RC2) · Anonymous Referee #2 · 13 Dec 2020

This article analyses the potential synoptic controls over central China during winter haze pollution episodes by using Lamb-Jenkension method and the NCEP/NCAR FNL operational global analysis data, and further evaluates the effectiveness of emission control to reduce PM2.5 under main synoptic conditions by GEOS-Chem model simulations. They found a substantial contribution of transportation in two synoptic patterns (SW-type and NW-type) and a dominated contribution of local emission sources in other two synoptic conditions (A-type and C-type). These results provide an opportunity to effectively mitigate haze pollution by local emission control actions in coordination with regional collaborative actions according to different synoptic patterns. The topic is of practical significance and the results are reliable. I would suggest for publication after addressing my comments below.

1. The present comparison and verification of control simulation results in GEOS-Chem is not enough. It can be further verified by using PM2.5 observation data in a larger region of China or component observations of PM2.5 at some specific sites.

2. The novelty of this study need to be further clarified. New understanding or improvement of conclusion and application or in methods should be provided to reflect the general interests of the work rather than the local interests.

3. Lines 105-109: several studies have investigated the potential effective emission reduction on ammonia, which should be reviewed here properly.

4. In Section 3.2, the mechanisms of heavy particle pollution caused by these four potential synoptic controls should be briefly discussed when describe characteristics of each synoptic pattern.

5. Lines 294-296: Why the four pollution episodes are selected?

6. Lines 304-308: The model control simulation is compared to PM2.5 observations at just one site (Jingzhou). Current comparison is insufficient to demonstrate the modeling performance.

7. Line 308-311: Model biases are generally attributed to resolution, emission errors, meteorology and chemical mechanism without statistical results of further sensitivity simulations. Be careful to discuss the model deviation.

8. Line 337: PSC -> PSCs

9. Line 359: The transportation of air pollutants from the south makes the proportion of the three
inorganic salts (45.7%) in Jingzhou area the smallest. Consider revising it like: The transport of air pollutants from the south leads to the smallest proportion of

the three inorganic salts (45.7%) in Jingzhou.

10. Line 482: remove potential synoptic controls or (PSC)

11. Line 494: contribute 82%/85% of PM2.5. Consider revising it like: dominate the contribution (82%/85%) to PM2.5.

---

## Author Response (AR1)

====================================================

Response to the First Referee

====================================================

Reviewer #1:

This paper by Yan et al. investigated the characteristics of winter haze episodes in Jingzhou of Central China under typical potential synoptic controls (PSCs) during November 2013-February 2014. Furthermore, they examined the contributions of local and transport of pollutants from surrounding regions to PM2.5 under different PSCs by applying the GEOS-Chem model with a high resolution. This work also studied the effectiveness of different emission control strategies in Jingzhou, Central China, and other surrounding regions under different PSCs, and highlights the importance of collaborative actions for PM2.5 mitigation under server haze pollution. In general, the study is well organized and worthy of publication. However, I have some specific comments that I feel deserve attention.

We thank the reviewer for comments, which have been incorporated to improve the manuscript.

Major comments

1. The writing should be improved before publication.

We thank the referee for this comment. We have made necessary corrections to grammar throughout the text (see details in the revision manuscript). We have polished the manuscript for all the authors.

2. The configuration of the model is vague. How many nested domains were applied in each simulation? What is the geographic coverage of each domain and the corresponding resolution? What are the emission inventories for each domain? A figure showing each nested domain is also highly recommended.

We thank the referee for this comment. We have added a figure in the revised file of supporting information to explain the geographic coverage of each domain and the corresponding resolution for GEOS_Chem global model ($2° \times 2.5°$, providing boundary condition to nested model) and nested model (70°E-140°E, 15°S-55°N; $0.25° \times 0.3125°$). The emission inventories for each domain are shown in the revised Table S1 and Table S2. We also revised the description in the text of Sect. 2.3.

[Figure]

Figure S3 The geographic coverage of each domain and the corresponding resolution for GEOS_Chem global model (2° × 2.5°) and nested model (70°E-140°E, 15°S-55°N; 0.25° × 0.3125°).

Table S1 Anthropogenic and natural source emission inventories adopted in the GEOS-Chem global modelling of this study

| Region | Abbreviation | Description | Resolution | Year | Species | Reference |
|---|---|---|---|---|---|---|
| Anthropogenic emission inventory | | | | | | |
| Global | EDGAR | EDGAR v4.2 anthropogenic + biofuel | 0.1°× 0.1°, monthly | 2013-2014 | NOx, $SO_2$, $SO_4^{2-}$, CO, $NH_3$ | http://edgar.jrc.ec.europa.eu/overview.php?v=42 |
| Global | BOND | BOND biofuel + anthropogenic BC + OC emissions | 1°×1°, monthly | 2000 | BC and OC | Bond et al. (2007) |
| Global | RETRO | RETRO anthropogenic + biofuel | 0.5°×0.5°, monthly | 2000 | NMVOCs[1] except $C_2H_6$ and $C_3H_8$ | ftp://ftp.retro.enes.org/pub/emissions/aggregated/anthro/0.5x0.5/2000/ |
| Global | SHIP | ICOADS ship emissions | 1°×1°, monthly | 2002 | $NO_x$, $SO_2$, CO | Wang et al. (2008) |
| Global | AEIC | Aircraft emissions | 1°×1°, monthly | 2005 | $NO_x$, $SO_2$, CO, NMVOCs[1], BC, OC | |
| China | MEIC | MEIC inventory for China | 0.25°×0.25°, monthly | 2013-2014 | $NO_x$, $SO_2$, CO, NMVOCs[1], $NH_3$ | http://www.meicmodel.org/. |
| USA | NEI2011 | US EPA NEI-2011 emission inventory | 0.1°× 0.1°, monthly | 2013-2014 | $NO_x$, $SO_2$, CO, NMVOCs[1], $NH_3$, BC, OC | https://www.epa.gov/air-emissions-inventories |
| Europe | EMEP | EMEP | 1°×1°, annual | 2013-2014 | $NO_x$, $SO_2$, CO | Auvray and Bey (2005) |
| Biomass burning emission inventory | | | | | | |
| Global | GFED4 | GFED4 biomass burning inventory | 0.25°× 0.25°, monthly | 2013-2014 | $NO_x$, $SO_2$, CO, NMVOCs, $NH_3$, BC, OC | http://www.globalfiredata.org, Giglio et al. (2013) |
| Biogenic emission inventory | | | | | | |

| Global | MEGAN | MEGAN v2.1 biogenic emissions | — | 2013-2014 | ISOP, monoterpenes, sesquiterpenes, MOH, ACET, ETOH, $CH_2O$, $ALD_2$, HCOOH, $C_2H_4$, TOLU, PRPE | Guenther et al. (2012) |
|---|---|---|---|---|---|---|
| Other natural emission inventory | | | | | | |
| Global | SoilNOx | Emission of $NO_x$ from soils and fertiliser use | — | 2013-2014 | NO | Hudman et al. (2012) |
| Global | LightNOx | $NO_x$ from lightning | — | 2013-2014 | NO | Murray et al. (2012) |

1. RETRO includes PRPE, $ALK_4$, $ALD_2$, $CH_2O$ and MEK; in the CTM, MEK emissions are further allocated to MEK (25 %) and ACET (75 %). AEIC and MEIC include PRPE, $C_2H_6$, $C_3H_8$, $ALK_4$, $ALD_2$, $CH_2O$, MEK and ACET. NEI2011 includes PRPE, $C_3H_8$, $ALK_4$, $CH_2O$, MEK and ACET. EMEP includes PRPE, $ALK_4$, $ALD_2$ and MEK. Emissions of $C_2H_6$ outside Asia are from Xiao et al. (2008).

Table S2 Anthropogenic and natural source emission inventories adopted in the GEOS-Chem nested modelling of this study

| Region | Abbreviation | Description | Resolution | Year | Species | Reference |
|---|---|---|---|---|---|---|
| Anthropogenic emission inventory | | | | | | |
| Non-China | EDGAR | EDGAR v4.2 anthropogenic + biofuel | 0.1°× 0.1°, monthly | 2013-2014 | NOx, $SO_2$, $SO_4^{2-}$, CO, $NH_3$ | http://edgar.jrc.ec.europa.eu/overview.php?v=42 |
| Nested domain | BOND | BOND biofuel + anthropogenic BC + OC emissions | 1°×1°, monthly | 2000 | BC and OC | Bond et al. (2007) |
| Non-China | RETRO | RETRO anthropogenic + biofuel | 0.5°×0.5°, monthly | 2000 | NMVOCs[1] except $C_2H_6$ and $C_3H_8$ | ftp://ftp.retro.enes.org/pub/emissions/aggregated/anthro/0.5x0.5/2000/ |
| Nested domain | SHIP | ICOADS ship emissions | 1°×1°, monthly | 2002 | $NO_x$, $SO_2$, CO | Wang et al. (2008) |

| | | | | | | |
|---|---|---|---|---|---|---|
| Nested domain | AEIC | Aircraft emissions | 1°×1°, monthly | 2005 | $NO_x$, $SO_2$, CO, NMVOCs[1], BC, OC | |
| China | MEIC | MEIC inventory for China | 0.25°×0.25°, monthly | 2013-2014 | $NO_x$, $SO_2$, CO, NMVOCs[1], $NH_3$ | http://www.meicmodel.org/. |
| Central China | SEEA | SEEA | 0.1°× 0.1°, monthly | 2017 | $NO_x$, $SO_2$, CO, $NH_3$, VOCs | |
| Biomass burning emission inventory | | | | | | |
| Nested domain | GFED4 | GFED4 biomass burning inventory | 0.25°× 0.25°, monthly | 2013-2014 | $NO_x$, $SO_2$, CO, NMVOCs, $NH_3$, BC, OC | http://www.globalfiredata.org, Giglio et al. (2013) |
| Biogenic emission inventory | | | | | | |
| Nested domain | MEGAN | MEGAN v2.1 biogenic emissions | — | 2013-2014 | ISOP, monoterpenes, sesquiterpenes, MOH, ACET, ETOH, $CH_2O$, $ALD_2$, HCOOH, $C_2H_4$, TOLU, PRPE | Guenther et al. (2012) |
| Other natural emission inventory | | | | | | |
| Nested domain | SoilNOx | Emission of $NO_x$ from soils and fertiliser use | — | 2013-2014 | NO | Hudman et al. (2012) |
| Nested domain | LightNOx | $NO_x$ from lightning | — | 2013-2014 | NO | Murray et al. (2012) |

1. RETRO includes PRPE, $ALK_4$, $ALD_2$, $CH_2O$ and MEK; in the CTM, MEK emissions are further allocated to MEK (25 %) and ACET (75 %). AEIC and MEIC include PRPE, $C_2H_6$, $C_3H_8$, $ALK_4$, $ALD_2$, $CH_2O$, MEK and ACET. NEI2011 includes PRPE, $C_3H_8$, $ALK_4$, $CH_2O$, MEK and ACET. EMEP includes PRPE, $ALK_4$, $ALD_2$ and MEK. Emissions of $C_2H_6$ outside Asia are from Xiao et al. (2008)

3. The circulation classification is the basis of all the analysis. Why did you choose the Lamb-Jenkension method? What are the advantages of this method compared to the ones used in other studies such as Chang and Zhan, 2017, Dai et al., 2021, etc.?

We thank the referee for this comment. We have reviewed the advantages of Lamb-Jenkension method with respect to the ones used in other studies in the revised Sect. 2.2: "Compared to the objective classification method PCA used in some studies (Chang and Zhan, 2017, Dai et al., 2021), this Lamb-Jenkension method is a combination of subjective and objective methods. After the objective judgment of the circulation, we also make subjective considerations to overcome the weaknesses of their respective, leading to better synoptic significance. Many works of circulation classification have used the Lamb-Jenkension method and reported that the analysis can well respond to the classification results (Philipp et al., 2016;Santurtun et al., 2015;Pope et al., 2015;Russo et al., 2014;Pope et al., 2014;Trigo and DaCamara, 2000)."

Philipp, A., Beck, C., Huth, R., and Jacobeit, J.: Development and comparison of circulation type classifications using the COST 733 dataset and software, International Journal of Climatology, 36, 2673-2691, 10.1002/joc.3920, 2016.

Pope, R. J., Savage, N. H., Chipperfield, M. P., Arnold, S. R., and Osborn, T. J.: The influence of synoptic weather regimes on UK air quality: analysis of satellite column $NO_2$, Atmospheric Science Letters, 15, 211-217, 10.1002/asl2.492, 2014.

Pope, R. J., Savage, N. H., Chipperfield, M. P., Ordonez, C., and Neal, L. S.: The influence of synoptic weather regimes on UK air quality: regional model studies of tropospheric column $NO_2$, Atmospheric Chemistry and Physics, 15, 11201-11215, 10.5194/acp-15-11201-2015, 2015.

Russo, A., Trigo, R. M., Martins, H., and Mendes, M. T.: $NO_2$, $PM_{10}$ and $O_3$ urban concentrations and its association with circulation weather types in Portugal, Atmospheric Environment, 89, 768-785, 10.1016/j.atmosenv.2014.02.010, 2014.

Santurtun, A., Carlos Gonzalez-Hidalgo, J., Sanchez-Lorenzo, A., and Teresa Zarrabeitia, M.: Surface ozone concentration trends and its relationship with weather types in Spain (2001-2010), Atmospheric Environment, 101, 10-22, 10.1016/j.atmosenv.2014.11.005, 2015.

Trigo, R. M., and DaCamara, C. C.: Circulation weather types and their influence on the precipitation regime in Portugal, International Journal of Climatology, 20, 1559-1581, 10.1002/1097-0088(20001115)20:13<1559::aid-joc555>3.0.co;2-5, 2000.

4. The validation of model performances is very weak. The bias of the modeled PM2.5 in Jingzhou can be as high as more than 100 μg/m3, what are the possible reasons? The authors simply claimed the uncertainties in emissions, meteorology, and chemistry might cause this discrepancy without any details. What are the amount of the PM2.5

precursors emitted in this study and how are the values compared to the published
literature? How about the meteorological parameters used by the model vs.
observations? The authors claimed an improvement in sulfate by the increase in
primarily emitted sulfate in the model, how is that compared with observations? They
also analyzed the changes in the chemical composition of PM2.5 under different typical
PSCs without examination of the model performances in the base case.

Thanks for this query and suggestion, which are valuable for us to improve this work.

In order to better evaluate the GEOS-Chem model performances, the spatial distribution
of $PM_{2.5}$ concentrations averaged over the four typical heavy pollution processes
simulated by the control (CON) simulation are compared with the observations (a total
of 633 sites) from Ministry of Ecology and Environment of China
(http://www.mee.gov.cn/) (revised Fig. 6). Similar to the underestimation in $PM_{2.5}$ at
Jingzhou, the underestimation is on a national scale when compared with the MEE
observations, with a bias of -29.3 μg/m$^3$, -18.7 μg/m$^3$, -39.0 μg/m$^3$ and -21.4 μg/m$^3$ on
average for SW-type, NW-type, A-type and C-type synoptic pattern, respectively (Fig.
6).

In order to explain the causes of the model discrepancy, we have added Table S3 to
show the observed (modeled) meteorological conditions averaged over these four
pollution episodes controlled by SW-type, NW-type, A-type and C-type synoptic
pattern, respectively. There is an overestimate in temperature and wind speed and an
underestimate in humidity, which can partly contribute to the underestimation of
modeled $PM_{2.5}$ concentrations. In addition, anthropogenic emissions for $PM_{2.5}$
precursors used here are for the year 2017 over Central China from our newly
developed SEEA inventory (Table S4). From 2013 to 2017, anthropogenic $NO_x$, $SO_2$,
and primary $PM_{2.5}$ emissions in Central China have declined substantially (Table S4),
due to the implementation of stringent emission control measures for the 12[th]-13[th] Five-
Year Plans (Zheng et al., 2018). The anthropogenic emissions biases may affect our
simulations and $PM_{2.5}$ attribution results to some extent.

We have no observations of the chemical compositions of $PM_{2.5}$. In order to examine
the model performances in the $PM_{2.5}$ chemical compositions, we have added Table 4 to
review the reported concentrations of $PM_{2.5}$ and the three inorganic salts (sulfate, nitrate
and ammonium) in other cities. The contributions of sulfate, nitrate and ammonium are
9.1%-31.9%, 5.7%-32.1% and 5.9%-13.3%, respectively. In the CON simulation, the
fractions of each inorganic salt to $PM_{2.5}$ for these four typical heavy pollution processes
are shown in revised Fig. S10, which are comparable to the previous results (Table 4).

[Figure]

Figure 6 Spatial distribution of observed (top row) and modeled (bottom row, by CON
case) PM$_{2.5}$ concentrations (μg/m$^3$) averaged over four severe pollution episodes
controlled by SW-type (first column), NW-type (second column), A-type (third
column) and C-type (forth column) synoptic pattern, respectively.

Table S3. The observed (modeled) meteorological conditions at Jingzhou averaged
over these four pollution episodes controlled by SW-type, NW-type, A-type and C-type
synoptic pattern, respectively.

| PSC | Temperature (°C) | Humidity (%) | Pressure (kpa) | Wind speed (m/s) |
|-----|------------------|--------------|----------------|------------------|
| SW  | 11.79 (12.96)    | 75.33 (69.25) | 1018.33 (1024.06) | 2.13 (3.09) |
| NW  | 3.61 (6.34)      | 71.16 (62.78) | 1027.53 (1031.53) | 1.44 (2.45) |
| A   | 5.81 (7.52)      | 64.96 (60.38) | 1026.63 (1028.66) | 1.45 (2.27) |
| C   | 9.60 (13.08)     | 78.10 (71.40) | 1011.48 (1014.24) | 1.88 (3.11) |

Table S4. The emission amount of PM$_{2.5}$ precursors over Central China calculated from

SEEA (for the year 2017) and MEIC (for the years of 2013, 2014 and 2017) inventory
(unit: $10^4$ ton).

| Category | $SO_2$ | $NO_X$ | $NH_3$ | $PM_{2.5}$ | CO | BC | OC | VOCs |
|---|---|---|---|---|---|---|---|---|
| SEEA (2017) | 48.4 | 94.0 | 54.6 | 26.4 | 553.8 | 6.2 | 12.9 | 117.2 |
| MEIC (2017) | 52.0 | 70.4 | 57.5 | 35.2 | 629.2 | 6.8 | 11.7 | 116.4 |
| MEIC (2013) | 173.3 | 98.4 | 62.4 | 54.5 | 836.5 | 9.2 | 16.7 | 116.6 |
| MEIC (2014) | 97.0 | 80.0 | 61.1 | 46.8 | 744.2 | 8.3 | 15.3 | 116.4 |

Table 4 The reported concentrations of $PM_{2.5}$ and the three inorganic salts (sulfate,
nitrate and ammonium, $\mu g/m^3$) in other cities.

[revised manuscript text omitted]

Minor comments:

Line 101-103: There must be many studies targeted the mitigation of PM2.5 at a
regional scale (Ding et al., 2019; Zhang et al., 2019, Xing et al., 2018, 2019; Fu et al.,
2017; etc.). Please rephrase this sentence.

We have rephrased this sentence: "Although there are many studies targeted PM$_{2.5}$
mitigations at a regional scale (Ding et al., 2019; Zhang et al., 2019, Xing et al., 2018,

2019; Fu et al., 2017; etc.), their results can not be directly applied to reduce winter
PM$_{2.5}$ pollution under various synoptic controls."

Line 148-150: It is very confusing. The circulation classification is based on the
meteorological data from November 2013 to February 2014, which is also the
simulation episode. Why did you use the hourly PM2.5 data from 2013-2018?

We used the hourly PM$_{2.5}$ from November 2013 to December 2018 to screen the pollution days (daily mean PM$_{2.5}$ larger than 150 μg/m$^3$) and applied the daily mean sea level pressure between 2013 and 2018 from the NCEP/NCAR FNL Operational Global

Analysis data to conduct the circulation classification. The meteorological observations at Jingzhou from November 2013 to February 2014 are used to analyze the meteorological characteristics during the period four severe particle pollution events occurred in succession over Central China. We have revised the Sect. 2.1:

"Hourly mass concentrations of PM$_{2.5}$ at Jingzhou (112.18°E, 30.33°N, 33.7 m)

from November 2013 to December 2018 are obtained from Hubei Environmental

Monitoring Central Station (http://sthjt.hubei.gov.cn/). We screen the pollution days with daily mean PM$_{2.5}$ concentrations larger than 150 μg/m$^3$ for circulation classification.

We use the daily mean sea level pressure (SLP) between 2013 and 2018 from the

National Centers for Environmental Prediction/National Center for Atmospheric

Research (NCEP/NCAR) Final (FNL) Operational Global Analysis data (horizontal resolution: 1° × 1°; temporal resolution: 6 hours; https://rda.ucar.edu/datasets/ds083.3/)

to conduct the classification of Lamb-Jenkension circulation types.

The meteorological data of surface observations at Jingzhou, including ambient temperature, relative humidity, wind speed, wind direction and atmospheric pressure, are obtained from Hubei Meteorological Information and Technology Support Center (http://hb.cma.gov.cn/qxfw/index.html). The data from November 2013 to February

2014 are used to analyze the meteorological characteristics during the period four severe particle pollution events occurred in succession over Central China (Fig. S1)."

Line 195: Did you do nested runs or just one domain covering China? Please make this clear.

We have specified the model setups in the revised sentences: "The nested model,
covering China (70°E-140°E, 15°S-55°N), is run with a horizontal resolution of 0.25°
latitude × 0.3125° longitude and 72 vertical layers. The boundary condition of nested
model is provided by the GEOS-Chem global model with a horizontal resolution of 2°
latitude × 2.5° longitude (Fig. S3). Both global and nested simulations, driven by the
GEOS-FP assimilated meteorological data, include detailed tropospheric Ozone-$NO_x$-
VOCs-$HO_x$-aerosol chemistry."

Line 205: The SEEA inventory was developed for the year 2017. Did you use it directly
without projection to the simulation episode? If you adjusted this inventory, what are
the factors applied for the PM2.5 precursors and how did you obtain those data?

Yes, we have used the SEEA inventory of the year 2017 directly without projection to
the simulation episode. The uncertainty discussion has been listed in Sect. 3.3:
"Anthropogenic emissions for $PM_{2.5}$ precursors used here are for the year 2017 over
Central China from SEEA inventory (Table S4). From 2013 to 2017, anthropogenic
$NO_x$, $SO_2$, and primary $PM_{2.5}$ emissions in Central China have declined substantially
(Table S4), due to implementation of stringent emission control measures for the 12th-
13th Five-Year Plans (Zheng et al., 2018). The anthropogenic emissions biases may
affect our simulations and $PM_{2.5}$ attribution results to some extent."

Line 215-217: Have you compared the modeled sulfate with observations, at least in
Jingzhou? How about the model performances of the other components of PM2.5?

We have no observations of the chemical compositions of $PM_{2.5}$. In order to examine
the model performances in the $PM_{2.5}$ chemical compositions, we have added Table 4 to
review the reported concentrations of $PM_{2.5}$ and the three inorganic salts (sulfate, nitrate
and ammonium) in other cities. The contributions of sulfate, nitrate and ammonium are
9.1%-31.9%, 5.7%-32.1% and 5.9%-13.3%, respectively. In the CON simulation, the
fractions of each inorganic salt to $PM_{2.5}$ for these four typical heavy pollution processes
are shown in revised Fig. S10, which are comparable to the previous observed results
(Table 4). Please see details in the response of major comment#4.

Line 305: Again, I am confused about the emissions used in the CON case. You listed
too many options for the anthropogenic source in Table S2. What inventories were
EXACTLY selected for the CON case? Did you do a global/regional nested run? Please
explain the choices of emissions in a separate column in the table.

We do a nested simulation, covering China (70°E-140°E, 15°S-55°N) with a horizontal
resolution of 0.25° latitude × 0.3125° longitude. The boundary condition of nested
model is provided by the GEOS-Chem global model with a horizontal resolution of 2°
latitude × 2.5° longitude (Fig. S3). The emission inventories for each domain are shown in the revised Table S1 and Table S2. Please see details in the response of major
comment#2.

Line 310: Please compare the meteorological field used in the model with observations
to confirm that statement. Also, there are no perfect mechanisms, inventories, or
parameterization of the model with no doubt. I suggest using "uncertainties".

We thank the referee for this comment. In order to explain the causes of the model
discrepancy, we have added Table S3 to show the observed (modeled) meteorological
conditions averaged over these four pollution episodes controlled by SW-type, NW-
type, A-type and C-type synoptic pattern, respectively. There is an overestimate in
temperature and wind speed and an underestimate in humidity, which can partly
contribute to the underestimation of modeled $PM_{2.5}$ concentrations. In addition,
anthropogenic emissions for $PM_{2.5}$ precursors used here are for the year 2017 over
Central China from SEEA inventory (Table S4). From 2013 to 2017, anthropogenic
$NO_x$, $SO_2$, and primary $PM_{2.5}$ emissions in Central China have declined substantially
(Table S4), due to implementation of stringent emission control measures for the 12th-
13th Five-Year Plans (Zheng et al., 2018). The anthropogenic emissions biases may
affect our simulations and $PM_{2.5}$ attribution results to some extent. Additionally, the
underestimation is on a national scale when compared with the MEE observations, with
a bias of -29.3 μg/m$^3$, -18.7 μg/m$^3$, -39.0 μg/m$^3$ and -21.4 μg/m$^3$ on average for SW-
type, NW-type, A-type and C-type synoptic pattern, respectively (Fig. 6). The national
negative biases may be also attributed to insufficient resolution of the model (Yan et
al., 2014) and imperfect chemical mechanisms (Yan et al., 2019). Please see details in
the response of major comment#4.

Line 323-324: A comparison of the modeled fractions of the inorganic salts to
observations, or reported values from other literature if no measurements are available.

We have no observations of the chemical compositions of $PM_{2.5}$. In order to examine
the model performances in the $PM_{2.5}$ chemical compositions, we have added Table 4 to
review the reported concentrations of $PM_{2.5}$ and the three inorganic salts (sulfate, nitrate
and ammonium) in other cities. The contributions of sulfate, nitrate and ammonium are
9.1%-31.9%, 5.7%-32.1% and 5.9%-13.3%, respectively. In the CON simulation, the
fractions of each inorganic salt to $PM_{2.5}$ for these four typical heavy pollution processes
are shown in revised Fig. S10, which are comparable to the previous observed results
(Table 4). Please see details in the response of major comment#4.

Line 324: "As shown in Table 3, ...."

Modified.

Line 358: How was this calculated? Please explain it.

We have added the explanation in the revised sentence: "In addition, the contributions from transboundary transport from non-Jingzhou Central China is simulated to be 12.0%
by comparing the results of XJ0 and XCC0."

Line 415-417: How about the contributions of transported pollutants to the chemical
composition of PM2.5 under the four PSCs?

We have discussed in the revised Sect. 3.4. During the pollution episodes of
transmission-pollution characteristics (SW/NW-type), the contribution of transported
pollutants to the chemical composition of $PM_{2.5}$ is significant. For the SW-type synoptic
controlled pollution event, the transport of air pollutants from the south leads to the
smallest proportion of the three inorganic salts (45.7%) in Jingzhou among the four
pollution episodes (50.3%-55.5% for other three episodes), because the emissions of
$SO_2$, $NO_2$ and $NH_3$ in the south (especially in Guangxi and Guizhou province) are
smaller than those in Central China (Li et al., 2017a). However, during the NW-type
synoptic controlled pollution episode, due to the transport contribution of pollutants
from northern China (with much higher anthropogenic emissions of $SO_2$, $NO_2$ and $NH_3$)
(Li et al., 2017a), the total proportion of the three inorganic salts is the highest (55.5%).
For the other two types (A/C-type) synoptic controlled pollutions, local emission
sources dominate the contributions and the contributions of transported pollutants to
the chemical composition of $PM_{2.5}$ are small.

Line 424: The base year of emission reduction is 2015 for the 13th Five-year plan,
which is quite different from your inventory. How effective is the designed reduction
ratio of the anthropogenic emissions in this study?

Although the base year of emission reduction is 2015 for the 13[th] Five-year plan, it does
not affect to use the simulation results of emission scenarios (with the reduction ratio
of 20% applied to the simulated year 2013/2014) to explore the emission reduction
effect of specific haze pollution events. We have added this illustration in the revised
Sect. 3.5.

Line 425 and 428-429: Please explain these abbreviations in the text as well.

We have revised these sentences as: "The differences in model results between CON
(control simulation) and JSN/JSNN/JALL (emissions of
$(SO_2+NO_x)/(SO_2+NO_x+NH_3)$/all pollution sources at Jingzhou are reduced by 20%)
represent the environmental benefits caused by different local emission reduction
scenarios. The potential $PM_{2.5}$ mitigations by joint prevention and control in different
regions are calculated by sensitivity experiments of CCALL (emissions of all pollution
sources over Central China are reduced by 20%), CNALL (over Central China and NCP
region), CPALL (over Central China and PRD region) and TALL (over Central China,
NCP, YRD, PRD and SCB regions)."

Line 437: I think an evaluation of the model performance in ammonium and/or ammonia is desired to confirm that.

We thank the referee for this comment. We have no observations of the chemical
compositions of $PM_{2.5}$. Thus we have removed this statement in the revised text.

Figure 6, 8, 9, 10: I suggest to show the fraction of each inorganic salt to PM2.5 rather
than their total mass.

We have shown the fraction of each inorganic salt to $PM_{2.5}$ in the revised Fig. S10.

[Figure]

Figure S10 Spatial distribution of $PM_{2.5}$ concentrations and the fraction of each inorganic salt (sulfate: second column; nitrate: third column; ammonium: forth column)

to $PM_{2.5}$ for these four typical heavy pollution processes simulated by GEOS-Chem control simulation.

Figure 11. It should be "TALL" in NW and C.

Modified.

We thank the referee for his/her reading of our manuscript. The comments and suggestions are valuable for us to improve our work.

In order to better evaluate the GEOS-Chem model performances, the spatial distribution of $PM_{2.5}$ concentrations averaged over the four typical heavy pollution processes simulated by the control (CON) simulation are compared with the observations (a total of 633 sites) from Ministry of Ecology and Environment of China (http://www.mee.gov.cn/) (revised Fig. 6). Similar to the underestimation in $PM_{2.5}$ at Jingzhou, the underestimation is on a national scale when compared with the MEE observations, with a bias of -29.3 $μg/m^3$, -18.7 $μg/m^3$, -39.0 $μg/m^3$ and -21.4 $μg/m^3$ on average for SW-type, NW-type, A-type and C-type synoptic pattern, respectively (Fig. 6).

We have no observations of the chemical compositions of $PM_{2.5}$. In order to examine the model performances in the $PM_{2.5}$ chemical compositions, we have added Table 4 to review the reported concentrations of $PM_{2.5}$ and the three inorganic salts (sulfate, nitrate and ammonium) in other cities. The contributions of sulfate, nitrate and ammonium are
9.1%-31.9%, 5.7%-32.1% and 5.9%-13.3%, respectively. In the CON simulation, the
fractions of each inorganic salt to PM$_{2.5}$ for these four typical heavy pollution processes
are shown in revised Fig. S10, which are comparable to the previous observed results
(Table 4).

[revised manuscript text omitted]

2. The novelty of this study need to be further clarified. New understanding or
improvement of conclusion and application or in methods should be provided to reflect
the general interests of the work rather than the local interests.

In Sect.1, we have further clarified and provided support that significant and new
scientific merits are presented in this work. The PM$_{2.5}$ pollution in China has been
continuously alleviating since 2013 as the implication of the Air Pollution Prevention
and Control Action Plan. However, severe particle pollution still occurs frequently in
autumn and winter, which is the major reason restricting the PM$_{2.5}$ to come up to
national standard. Currently, how to effectively reduce emissions in autumn and winter
is the key to mitigate haze pollution in China. Previous studies have highlighted that
different levels of PM$_{2.5}$ pollutions are closely related to the dominant synoptic patterns
in different regions, and they attribute the large spatial variability of pollution to the regional transport contributions, not only the different local sources of $PM_{2.5}$. Thus, heavy pollution prevention and control needs to consider the weather situation, otherwise local emission reduction measures would not work well. However, under different synoptic conditions, how to effectively reduce local and regional emissions to control haze pollution is rarely reported. In order to investigate the effectiveness of emission control to reduce $PM_{2.5}$ pollution under various potential synoptic controls, we take the severe particle pollution of winter haze episodes over Central China with transmission-pollution characteristics as an example. This study combines the atmospheric (circulation classification) and environmental (chemical transport modeling) research methods and could provide reference for emission control of severe winter haze pollution under different weather types, and provide basis for regional air quality policy-making.

3. Lines 105-109: several studies have investigated the potential effective emission reduction on ammonia, which should be reviewed here properly.

We have added the review of studies on potential efficiency of ammonia emission reduction in alleviating particulate pollution: "Moreover, current emission reduction policies in China mainly aimed at sulfur dioxide ($SO_2$) and nitrogen dioxide ($NO_2$), ignoring the effective emission reduction on ammonia ($NH_3$), although some modeling works have discussed the effectiveness of ammonia emission reduction for $PM_{2.5}$ mitigations (Liu et al., 2019; Ye et al., 2019; Xu et al., 2019; Bai et al., 2019)."

Bai, Z., Winiwarter, W., Klimont, Z., Velthof, G., Misselbrook, T., Zhao, Z., Jin, X., Oenema, O., Hu, C., and Ma, L.: Further Improvement of Air Quality in China Needs Clear Ammonia Mitigation Target, Environmental Science & Technology, 53, 10542-10544, 10.1021/acs.est.9b04725, 2019.

Liu, M., Huang, X., Song, Y., Tang, J., Cao, J., Zhang, X., Zhang, Q., Wang, S., Xu, T., Kang, L., Cai, X., Zhang, H., Yang, F., Wang, H., Yu, J. Z., Lau, A. K. H., He, L., Huang, X., Duan, L., Ding, A., Xue, L., Gao, J., Liu, B., and Zhu, T.: Ammonia emission control in China would mitigate haze pollution and nitrogen deposition, but worsen acid rain, Proceedings of the National Academy of Sciences of the United States of America, 116, 7760-7765, 10.1073/pnas.1814880116, 2019.

Xu, Z., Liu, M., Zhang, M., Song, Y., Wang, S., Zhang, L., Xu, T., Wang, T., Yan, C., Zhou, T., Sun, Y., Pan, Y., Hu, M., Zheng, M., and Zhu, T.: High efficiency of livestock ammonia emission controls in alleviating particulate nitrate during a severe winter haze episode in northern China, Atmospheric Chemistry and Physics, 19, 5605-5613, 10.5194/acp-19-5605-2019, 2019.

Ye, Z., Guo, X., Cheng, L., Cheng, S., Chen, D., Wang, W., and Liu, B.: Reducing $PM_{2.5}$ and secondary inorganic aerosols by agricultural ammonia emission mitigation within the Beijing-Tianjin-Hebei region, China, Atmospheric Environment, 219,

10.1016/j.atmosenv.2019.116989, 2019.

4. In Section 3.2, the mechanisms of heavy particle pollution caused by these four
potential synoptic controls should be briefly discussed when describe characteristics of
each synoptic pattern.

We have briefly discussed the mechanisms of heavy particle pollution caused by the
four PSCs in the revised Section 3.2, when describe characteristics of each synoptic
pattern:

[revised manuscript text omitted]

5. Lines 294-296: Why the four pollution episodes are selected?

We have explained in the revised sentences: "In order to reduce the simulation cost, the continuous four severe haze episodes occurred during November, 2013-February, 2014 are selected. These four haze episodes are controlled by the synoptic pattern of SW-type (18-25 November, 2013), NW-type (19-26 December, 2013), A-type (14-21 January, 2014) and C-type (26 January - 2 February, 2014), respectively."

6. Lines 304-308: The model control simulation is compared to PM2.5 observations at just one site (Jingzhou). Current comparison is insufficient to demonstrate the modeling performance.

In order to better evaluate the GEOS-Chem model performances, the spatial distribution of $PM_{2.5}$ concentrations averaged over the four typical heavy pollution processes simulated by the control (CON) simulation are compared with the observations (a total of 633 sites) from Ministry of Ecology and Environment of China (http://www.mee.gov.cn/) (revised Fig. 6). Please see details in the response of comment#1.

7. Line 308-311: Model biases are generally attributed to resolution, emission errors,
meteorology and chemical mechanism without statistical results of further sensitivity
simulations. Be careful to discuss the model deviation.

In order to explain the causes of the model discrepancy, we have added Table S3 to
show the observed (modeled) meteorological conditions averaged over these four
pollution episodes controlled by SW-type, NW-type, A-type and C-type synoptic
pattern, respectively. There is an overestimate in temperature and wind speed and an
underestimate in humidity, which can partly contribute to the underestimation of
modeled $PM_{2.5}$ concentrations. In addition, anthropogenic emissions for $PM_{2.5}$
precursors used here are for the year 2017 over Central China from SEEA inventory
(revised Table S4). From 2013 to 2017, anthropogenic $NO_x$, $SO_2$, and primary $PM_{2.5}$
emissions in Central China have declined substantially (revised Table S4), due to the
implementation of stringent emission control measures for the 12th-13th Five-Year Plans
(Zheng et al., 2018). The anthropogenic emissions biases may affect our simulations
and $PM_{2.5}$ attribution results to some extent. Additionally, the underestimation is on a
national scale when compared with the MEE observations, with a bias of -29.3 $\mu g/m^3$,
-18.7 $\mu g/m^3$, -39.0 $\mu g/m^3$ and -21.4 $\mu g/m^3$ on average for SW-type, NW-type, A-type
and C-type synoptic pattern, respectively (revised Fig. 6, see figure in the response of
comment#1). The national negative biases may be also attributed to insufficient
resolution of the model (Yan et al., 2014) and imperfect chemical mechanisms (Yan et
al., 2019).

Table S3. The observed (modeled) meteorological conditions at Jingzhou averaged
over these four pollution episodes controlled by SW-type, NW-type, A-type and C-type
synoptic pattern, respectively.

| PSC | Temperature (°C) | Humidity (%) | Pressure (kpa) | Wind speed (m/s) |
| --- | --- | --- | --- | --- |
| SW | 11.79 (12.96) | 75.33 (69.25) | 1018.33 (1024.06) | 2.13 (3.09) |
| NW | 3.61 (6.34) | 71.16 (62.78) | 1027.53 (1031.53) | 1.44 (2.45) |
| A | 5.81 (7.52) | 64.96 (60.38) | 1026.63 (1028.66) | 1.45 (2.27) |
| C | 9.60 (13.08) | 78.10 (71.40) | 1011.48 (1014.24) | 1.88 (3.11) |

Table S4. The emission amount of $PM_{2.5}$ precursors over Central China calculated from
SEEA (for the year 2017) and MEIC (for the years of 2013, 2014 and 2017) inventory
(unit: $10^4$ ton).

| Category | SO$_2$ | NO$_X$ | NH$_3$ | PM$_{2.5}$ | CO | BC | OC | VOCs |
|---|---|---|---|---|---|---|---|---|
| SEEA (2017) | 48.4 | 94.0 | 54.6 | 26.4 | 553.8 | 6.2 | 12.9 | 117.2 |
| MEIC (2017) | 52.0 | 70.4 | 57.5 | 35.2 | 629.2 | 6.8 | 11.7 | 116.4 |
| MEIC (2013) | 173.3 | 98.4 | 62.4 | 54.5 | 836.5 | 9.2 | 16.7 | 116.6 |
| MEIC (2014) | 97.0 | 80.0 | 61.1 | 46.8 | 744.2 | 8.3 | 15.3 | 116.4 |

8. Line 337: PSC -> PSCs

Modified.

9. Line 359: The transportation of air pollutants from the south makes the proportion of
the three inorganic salts (45.7%) in Jingzhou area the smallest. Consider revising it like:
The transport of air pollutants from the south leads to the smallest proportion of the
three inorganic salts (45.7%) in Jingzhou.

Modified.

10. Line 482: remove potential synoptic controls or (PSC)

Modified.

11. Line 494: contribute 82%/85% of PM2.5. Consider revising it like: dominate the
contribution (82%/85%) to PM2.5.

Modified.

[revised manuscript text omitted]

Font color: Text 1

| Page 77: [2] Formatted | Microsoft Office User | 1/3/21 12:42 PM |
| --- | --- | --- |

Font color: Text 1

| Page 77: [3] Formatted | Microsoft Office User | 1/3/21 12:42 PM |
| --- | --- | --- |

Font color: Text 1

| Page 77: [3] Formatted | Microsoft Office User | 1/3/21 12:42 PM |
| --- | --- | --- |

Font color: Text 1

| Page 77: [4] Formatted | Microsoft Office User | 1/3/21 12:42 PM |
| --- | --- | --- |

Font color: Text 1

| Page 77: [4] Formatted | Microsoft Office User | 1/3/21 12:42 PM |
| --- | --- | --- |

Font color: Text 1

| Page 77: [5] Formatted | Microsoft Office User | 1/3/21 12:42 PM |
| --- | --- | --- |

Font color: Text 1

| Page 77: [5] Formatted | Microsoft Office User | 1/3/21 12:42 PM |
| --- | --- | --- |

Font color: Text 1

| Page 77: [6] Formatted | Microsoft Office User | 1/3/21 12:42 PM |
| --- | --- | --- |

Font color: Text 1

| Page 77: [6] Formatted | Microsoft Office User | 1/3/21 12:42 PM |
| --- | --- | --- |

Font color: Text 1

| Page 77: [7] Formatted | Microsoft Office User | 1/3/21 12:42 PM |
| --- | --- | --- |

Font color: Text 1

| Page 77: [8] Formatted | Microsoft Office User | 1/3/21 12:42 PM |
| --- | --- | --- |

Font color: Text 1

| Page 77: [8] Formatted | Microsoft Office User | 1/3/21 12:42 PM |
| --- | --- | --- |

Font color: Text 1

| Page 77: [9] Formatted | Microsoft Office User | 1/3/21 12:42 PM |
| --- | --- | --- |

Font color: Text 1

| Page 77: [9] Formatted | Microsoft Office User | 1/3/21 12:42 PM |
| --- | --- | --- |

Font color: Text 1

| Page 77: [10] Formatted | Microsoft Office User | 1/3/21 12:42 PM |
| --- | --- | --- |

Font color: Text 1

| Page 77: [10] Formatted | Microsoft Office User | 1/3/21 12:42 PM |
| --- | --- | --- |

Font color: Text 1

| Page 77: [11] Formatted | Microsoft Office User | 1/3/21 12:42 PM |
| --- | --- | --- |

Font color: Text 1

| Page 77: [11] Formatted | Microsoft Office User | 1/3/21 12:42 PM |
| --- | --- | --- |

Font color: Text 1

| Page 77: [12] Formatted | Microsoft Office User | 1/3/21 12:42 PM |
|---|---|---|

Font color: Text 1

| Page 77: [13] Formatted | Microsoft Office User | 1/3/21 12:42 PM |
|---|---|---|

Font color: Text 1

| Page 77: [13] Formatted | Microsoft Office User | 1/3/21 12:42 PM |
|---|---|---|

Font color: Text 1

| Page 77: [14] Formatted | Microsoft Office User | 1/3/21 12:42 PM |
|---|---|---|

Font color: Text 1

| Page 77: [14] Formatted | Microsoft Office User | 1/3/21 12:42 PM |
|---|---|---|

Font color: Text 1

| Page 77: [15] Formatted | Microsoft Office User | 1/3/21 12:42 PM |
|---|---|---|

Font color: Text 1

| Page 77: [15] Formatted | Microsoft Office User | 1/3/21 12:42 PM |
|---|---|---|

Font color: Text 1

| Page 77: [16] Formatted | Microsoft Office User | 1/3/21 12:42 PM |
|---|---|---|

Font color: Text 1

| Page 77: [16] Formatted | Microsoft Office User | 1/3/21 12:42 PM |
|---|---|---|

Font color: Text 1

| Page 77: [17] Formatted | Microsoft Office User | 1/3/21 12:42 PM |
|---|---|---|

Font color: Text 1

| Page 77: [18] Formatted | Microsoft Office User | 1/3/21 12:42 PM |
|---|---|---|

Font color: Text 1

| Page 77: [18] Formatted | Microsoft Office User | 1/3/21 12:42 PM |
|---|---|---|

Font color: Text 1

| Page 77: [19] Formatted | Microsoft Office User | 1/3/21 12:42 PM |
|---|---|---|

Font color: Text 1

| Page 77: [19] Formatted | Microsoft Office User | 1/3/21 12:42 PM |
|---|---|---|

Font color: Text 1

| Page 77: [20] Formatted | Microsoft Office User | 1/3/21 12:42 PM |
|---|---|---|

Font color: Text 1

| Page 77: [20] Formatted | Microsoft Office User | 1/3/21 12:42 PM |
|---|---|---|

Font color: Text 1

| Page 77: [21] Formatted | Microsoft Office User | 1/3/21 12:42 PM |
|---|---|---|

Font color: Text 1

| Page 77: [21] Formatted | Microsoft Office User | 1/3/21 12:42 PM |
|---|---|---|

Font color: Text 1

| Page 77: [22] Formatted | Microsoft Office User | 1/3/21 12:42 PM |
|---|---|---|

Font color: Text 1

| Page 77: [23] Formatted | Microsoft Office User | 1/3/21 12:42 PM |
|---|---|---|

Font color: Text 1

| Page 77: [23] Formatted | Microsoft Office User | 1/3/21 12:42 PM |
|---|---|---|

Font color: Text 1

| Page 77: [24] Formatted | Microsoft Office User | 1/3/21 12:42 PM |
|---|---|---|

Font color: Text 1

| Page 77: [24] Formatted | Microsoft Office User | 1/3/21 12:42 PM |
|---|---|---|

Font color: Text 1

| Page 77: [25] Formatted | Microsoft Office User | 1/3/21 12:42 PM |
|---|---|---|

Font color: Text 1

| Page 77: [25] Formatted | Microsoft Office User | 1/3/21 12:42 PM |
|---|---|---|

Font color: Text 1

| Page 77: [26] Formatted | Microsoft Office User | 1/3/21 12:42 PM |
|---|---|---|

Font color: Text 1

| Page 77: [26] Formatted | Microsoft Office User | 1/3/21 12:42 PM |
|---|---|---|

Font color: Text 1

| Page 77: [27] Formatted | Microsoft Office User | 1/3/21 12:42 PM |
|---|---|---|

Font color: Text 1

| Page 77: [28] Formatted | Microsoft Office User | 1/3/21 12:42 PM |
|---|---|---|

Font color: Text 1

| Page 77: [28] Formatted | Microsoft Office User | 1/3/21 12:42 PM |
|---|---|---|

Font color: Text 1

| Page 77: [29] Formatted | Microsoft Office User | 1/3/21 12:42 PM |
|---|---|---|

Font color: Text 1

| Page 77: [29] Formatted | Microsoft Office User | 1/3/21 12:42 PM |
|---|---|---|

Font color: Text 1

| Page 77: [30] Formatted | Microsoft Office User | 1/3/21 12:42 PM |
|---|---|---|

Font color: Text 1

| Page 77: [30] Formatted | Microsoft Office User | 1/3/21 12:42 PM |
|---|---|---|

Font color: Text 1

| Page 77: [31] Formatted | Microsoft Office User | 1/3/21 12:42 PM |
|---|---|---|

Font color: Text 1

| Page 77: [31] Formatted | Microsoft Office User | 1/3/21 12:42 PM |
|---|---|---|

Font color: Text 1

| Page 77: [32] Formatted | Microsoft Office User | 1/3/21 12:42 PM |
|---|---|---|

Font color: Text 1

| Page 77: [33] Formatted | Microsoft Office User | 1/3/21 12:42 PM |
|---|---|---|

Font color: Text 1

| Page 77: [33] Formatted | Microsoft Office User | 1/3/21 12:42 PM |
|---|---|---|

Font color: Text 1

| Page 77: [34] Formatted | Microsoft Office User | 1/3/21 12:42 PM |
|---|---|---|

Font color: Text 1

| Page 77: [34] Formatted | Microsoft Office User | 1/3/21 12:42 PM |
|---|---|---|

Font color: Text 1

| Page 77: [35] Formatted | Microsoft Office User | 1/3/21 12:42 PM |
|---|---|---|

Font color: Text 1

| Page 77: [35] Formatted | Microsoft Office User | 1/3/21 12:42 PM |
|---|---|---|

Font color: Text 1

| Page 77: [36] Formatted | Microsoft Office User | 1/3/21 12:42 PM |
|---|---|---|

Font color: Text 1

| Page 77: [36] Formatted | Microsoft Office User | 1/3/21 12:42 PM |
|---|---|---|

Font color: Text 1

| Page 77: [37] Formatted | Microsoft Office User | 1/3/21 12:42 PM |
|---|---|---|

Font color: Text 1

| Page 77: [38] Formatted | Microsoft Office User | 1/3/21 12:42 PM |
|---|---|---|

Font color: Text 1

| Page 77: [38] Formatted | Microsoft Office User | 1/3/21 12:42 PM |
|---|---|---|

Font color: Text 1

| Page 77: [39] Formatted | Microsoft Office User | 1/3/21 12:42 PM |
|---|---|---|

Font color: Text 1

| Page 77: [39] Formatted | Microsoft Office User | 1/3/21 12:42 PM |
|---|---|---|

Font color: Text 1

| Page 77: [40] Formatted | Microsoft Office User | 1/3/21 12:42 PM |
|---|---|---|

Font color: Text 1

| Page 77: [40] Formatted | Microsoft Office User | 1/3/21 12:42 PM |
|---|---|---|

Font color: Text 1

| Page 77: [41] Formatted | Microsoft Office User | 1/3/21 12:42 PM |
|---|---|---|

Font color: Text 1

| Page 77: [41] Formatted | Microsoft Office User | 1/3/21 12:42 PM |
|---|---|---|

Font color: Text 1

| Page 77: [42] Formatted | Microsoft Office User | 1/3/21 12:42 PM |
|---|---|---|

Font color: Text 1

| Page 77: [43] Formatted | Microsoft Office User | 1/3/21 12:42 PM |
|---|---|---|

Font color: Text 1

| Page 77: [44] Formatted | Microsoft Office User | 1/3/21 12:42 PM |
|---|---|---|

Font color: Text 1

| Page 77: [45] Formatted | Microsoft Office User | 1/3/21 12:42 PM |
|---|---|---|

Font color: Text 1

| Page 77: [46] Formatted | Microsoft Office User | 1/3/21 12:42 PM |
|---|---|---|

Font color: Text 1

| Page 77: [47] Formatted | Microsoft Office User | 1/3/21 12:42 PM |
|---|---|---|

Font color: Text 1

| Page 77: [48] Formatted | Microsoft Office User | 1/3/21 12:42 PM |
|---|---|---|

Font color: Text 1

| Page 77: [49] Formatted | Microsoft Office User | 1/3/21 12:42 PM |
|---|---|---|

Font color: Text 1

| Page 77: [50] Formatted | Microsoft Office User | 1/3/21 12:42 PM |
|---|---|---|

Font color: Text 1

| Page 77: [51] Formatted | Microsoft Office User | 1/3/21 12:42 PM |
|---|---|---|

Font color: Text 1

| Page 77: [52] Formatted | Microsoft Office User | 1/3/21 12:42 PM |
|---|---|---|

Font color: Text 1

| Page 77: [53] Formatted | Microsoft Office User | 1/3/21 12:42 PM |
|---|---|---|

Font color: Text 1

| Page 77: [54] Formatted | Microsoft Office User | 1/3/21 12:42 PM |
|---|---|---|

Font color: Text 1

| Page 77: [54] Formatted | Microsoft Office User | 1/3/21 12:42 PM |

Font color: Text 1

| Page 77: [55] Formatted | Microsoft Office User | 1/3/21 12:42 PM |

Font color: Text 1

| Page 77: [55] Formatted | Microsoft Office User | 1/3/21 12:42 PM |

Font color: Text 1

| Page 77: [56] Formatted | Microsoft Office User | 1/3/21 12:42 PM |

Font color: Text 1

| Page 77: [56] Formatted | Microsoft Office User | 1/3/21 12:42 PM |

Font color: Text 1

| Page 77: [57] Formatted | Microsoft Office User | 1/3/21 12:42 PM |

Font color: Text 1

| Page 77: [57] Formatted | Microsoft Office User | 1/3/21 12:42 PM |

Font color: Text 1

| Page 77: [58] Formatted | Microsoft Office User | 1/3/21 12:42 PM |

Font color: Text 1

| Page 77: [59] Formatted | Microsoft Office User | 1/3/21 12:42 PM |

Font color: Text 1

| Page 77: [59] Formatted | Microsoft Office User | 1/3/21 12:42 PM |

Font color: Text 1

| Page 77: [60] Formatted | Microsoft Office User | 1/3/21 12:42 PM |

Font color: Text 1

| Page 77: [60] Formatted | Microsoft Office User | 1/3/21 12:42 PM |

Font color: Text 1

| Page 77: [61] Formatted | Microsoft Office User | 1/3/21 12:42 PM |

Font color: Text 1

| Page 77: [61] Formatted | Microsoft Office User | 1/3/21 12:42 PM |

Font color: Text 1

| Page 77: [62] Formatted | Microsoft Office User | 1/3/21 12:42 PM |

Font color: Text 1

| Page 77: [62] Formatted | Microsoft Office User | 1/3/21 12:42 PM |

Font color: Text 1

| Page 78: [63] Formatted | Microsoft Office User | 1/3/21 12:42 PM |

Font color: Text 1

| Page 78: [63] Formatted | Microsoft Office User | 1/3/21 12:42 PM |

Font color: Text 1

| Page 78: [64] Formatted | Microsoft Office User | 1/3/21 12:42 PM |

Font color: Text 1

| Page 78: [64] Formatted | Microsoft Office User | 1/3/21 12:42 PM |

Font color: Text 1

| Page 78: [65] Formatted | Microsoft Office User | 1/3/21 12:42 PM |

Font color: Text 1

| Page 78: [65] Formatted | Microsoft Office User | 1/3/21 12:42 PM |

Font color: Text 1

| Page 78: [66] Formatted | Microsoft Office User | 1/3/21 12:42 PM |

Font color: Text 1

| Page 78: [66] Formatted | Microsoft Office User | 1/3/21 12:42 PM |

Font color: Text 1

| Page 78: [67] Formatted | Microsoft Office User | 1/3/21 12:42 PM |

Font color: Text 1

| Page 78: [67] Formatted | Microsoft Office User | 1/3/21 12:42 PM |

Font color: Text 1

| Page 78: [68] Formatted | Microsoft Office User | 1/3/21 12:42 PM |

Font color: Text 1

| Page 78: [68] Formatted | Microsoft Office User | 1/3/21 12:42 PM |

Font color: Text 1

| Page 78: [69] Formatted | Microsoft Office User | 1/3/21 12:42 PM |

Font color: Text 1

| Page 78: [69] Formatted | Microsoft Office User | 1/3/21 12:42 PM |

Font color: Text 1

| Page 78: [70] Formatted | Microsoft Office User | 1/3/21 12:42 PM |

Font color: Text 1

| Page 78: [70] Formatted | Microsoft Office User | 1/3/21 12:42 PM |

Font color: Text 1

| Page 78: [71] Formatted | Microsoft Office User | 1/3/21 12:42 PM |

Font color: Text 1

| Page 78: [71] Formatted | Microsoft Office User | 1/3/21 12:42 PM |

Font color: Text 1

| Page 78: [72] Formatted | Microsoft Office User | 1/3/21 12:42 PM |

Font color: Text 1

| Page 78: [72] Formatted | Microsoft Office User | 1/3/21 12:42 PM |

Font color: Text 1

| Page 78: [73] Formatted | Microsoft Office User | 1/3/21 12:42 PM |

Font color: Text 1

| Page 78: [73] Formatted | Microsoft Office User | 1/3/21 12:42 PM |

Font color: Text 1

| Page 78: [74] Formatted | Microsoft Office User | 1/3/21 12:42 PM |

Font color: Text 1

| Page 78: [74] Formatted | Microsoft Office User | 1/3/21 12:42 PM |

Font color: Text 1

| Page 78: [75] Formatted | Microsoft Office User | 1/3/21 12:42 PM |

Font color: Text 1

| Page 78: [75] Formatted | Microsoft Office User | 1/3/21 12:42 PM |

Font color: Text 1

| Page 78: [76] Formatted | Microsoft Office User | 1/3/21 12:42 PM |

Font color: Text 1

| Page 78: [76] Formatted | Microsoft Office User | 1/3/21 12:42 PM |

Font color: Text 1

| Page 78: [77] Formatted | Microsoft Office User | 1/3/21 12:42 PM |

Font color: Text 1

| Page 78: [77] Formatted | Microsoft Office User | 1/3/21 12:42 PM |

Font color: Text 1

| Page 78: [78] Formatted | Microsoft Office User | 1/3/21 12:42 PM |

Font color: Text 1

| Page 78: [78] Formatted | Microsoft Office User | 1/3/21 12:42 PM |

Font color: Text 1

| Page 78: [79] Formatted | Microsoft Office User | 1/3/21 12:42 PM |

Font color: Text 1

| Page 78: [79] Formatted | Microsoft Office User | 1/3/21 12:42 PM |

Font color: Text 1

| Page 78: [80] Formatted | Microsoft Office User | 1/3/21 12:42 PM |

Font color: Text 1

| Page 78: [80] Formatted | Microsoft Office User | 1/3/21 12:42 PM |

Font color: Text 1

| Page 78: [81] Formatted | Microsoft Office User | 1/3/21 12:42 PM |

Font color: Text 1

| Page 78: [81] Formatted | Microsoft Office User | 1/3/21 12:42 PM |

Font color: Text 1

| Page 78: [82] Formatted | Microsoft Office User | 1/3/21 12:42 PM |

Font color: Text 1

| Page 78: [82] Formatted | Microsoft Office User | 1/3/21 12:42 PM |

Font color: Text 1